# PSO-Based Target Localization and Tracking in Wireless Sensor Networks

**Shu-Hung Lee** [1], **Chia-Hsin Cheng** [2,*] , **Chien-Chih Lin** [2] **and Yung-Fa Huang** [3,*]

1    School of Intelligent Manufacturing and Automotive Engineering, Guangdong Business and Technology University, Zhaoqing 526020, China
2    Department of Electrical Engineering, National Formosa University, Yunlin 632301, Taiwan
3    Department of Information and Communication Engineering, Chaoyang University of Technology, Taichung 413310, Taiwan
*    Correspondence: chcheng@nfu.edu.tw (C.-H.C.); yfahuang@cyut.edu.tw (Y.-F.H.); Tel.: +886-4-2332-3000 (Y.-F.H.)

**Abstract:** Research of target localization and tracking is always a remarkable problem in the application of wireless sensor networks (WSNs) technology. There are many kinds of research and applications of target localization and tracking, such as Angle of Arrival (AOA), Time of Arrival (TOA), and Time Difference of Arrival (TDOA). The target localization accuracy for TOA, TDOA, and AOA is better than RSS. However, the required devices in the TOA, TDOA, and AOA are more expensive than RSS. In addition, the computational complexity of TOA, TDOA, and AOA is also more complicated than RSS. This paper uses a particle swarm optimization (PSO) algorithm with the received signal strength index (RSSI) channel model for indoor target localization and tracking. The performance of eight different method combinations of random or regular points, fixed or adaptive weights, and the region segmentation method (RSM) proposed in this paper for target localization and tracking is investigated for the number of particles in the PSO algorithm with 12, 24, 52, 72, and 100. The simulation results show that the proposed RSM method can reduce the number of particles used in the PSO algorithm and improve the speed of positioning and tracking without affecting the accuracy of target localization and tracking. The total average localization time for target localization and tracking with the RSM method can be reduced by 48.95% and 34.14%, respectively, and the average accuracy of target tracking reaches up to 93.09%.

**Keywords:** WSNs; target localization; target tracking; PSO; RSSI

## 1. Introduction

With the development of science and technology, the way of human communication has evolved from wired to wireless, which has also prompted many applications of wireless communication. Wireless Sensor Networks (WSNs) are one of the most important in wireless communication applications. WSNs are composed of multiple sensor nodes, which collect different environmental information such as temperature, sound, or pressure data and then process the data according to a variety of applications. WSNs are applied in many fields such as military monitoring, environmental detection, smart homes, and target localization and tracking [1,2]. The wireless ad hoc network composes multiple mobile hosts, such as notebook computers or smartphones. Each mobile host is equipped with a wireless network card so that each device can communicate with the other. An ad hoc network is characterized in that there is no base station in the network, and the transmission work is connected peer-to-peer between networks. Its advantage is that the network can be quickly formed, and it has considerable convenience and adaptability. Compared with the wireless networks of the general base station structure, such as GSM, GPRS, or 4G and other systems, the deployment cost of the wireless ad hoc network is greatly reduced. A wireless ad hoc network is the predecessor of the wireless sensor network; both are

infrastructure-less network forms. They have much in common in terms of concept, but the construction conditions of wireless sensor networks are more stringent. The wireless sensor network originated from the Smart Dust project at the University of California, Berkeley. Its purpose was to develop a set of tiny sensor nodes for the U.S. Department of Defense and apply it to military-related intelligence collection. Under the wireless network architecture, the design of the sensor is to save power and be small as the primary goal. Sensors must not only have sensing capabilities, but also have communication capabilities. Each sensor is like a tiny computer. Since then, the University of California, Los Angeles, and the Massachusetts Institute of Technology have successively developed different sensors to expand the application of wireless sensor network.

This paper focuses on the application of wireless sensor networks in target localization and tracking. Generally speaking, the positioning methods of wireless sensor networks can be divided into range-based and range-free [1,2]. The former is most representative of a Global Positioning System (GPS). The advantage of GPS is that it has high positioning accuracy, but its disadvantages are that the construction cost is high and it is easily affected by factors such as weather effects or building shelters, which will affect the range of use and positioning quality. The former is most represented by the GPS. In practical applications, to meet the needs of dynamic events, the sensor needs to have the ability to move [3,4]. Therefore, this study will explore how to use the characteristics of information sharing and the movement of elements or particles (sensors) in artificial intelligence algorithms, combined with the received signal strength index (RSSI) channel model to improve the performance of target positioning and tracking in wireless sensor networks.

In the target localization and tracking system, each sensor needs to know its position, and the fastest way is to use GPS for target localization and tracking. However, it is very expensive to install a GPS positioning system on large-scale sensing equipment, so many target localization and tracking methods of wireless sensor networks have been proposed, such as Received Signal Strength (RSS), Time of Arrival (TOA), Time Difference of Arrival (TDOA), Angle of Arrival (AOA), etc. [5–8]. The sensors communicate with each other to receive signals to the target point to estimate the target position. However, in these methods, the equipment required by TOA, TDOA, and AOA is more expensive than RSS, and the computational complexity is higher. Therefore, this paper uses RSS to estimate the distance between the target point and the algorithm (sensor). However, the shortcoming of RSS is that it is easily affected by the change in environment, which will cause the estimation error of the target position to be too large. Therefore, reducing the positioning error caused by the fluctuation of RSS is one of the issues to be discussed in this study.

A variety of indoor localization technologies have been proposed in the literature. Localization techniques on signal processing methodology, such as proximity sensing, lateration, angulation, dead reckoning, fingerprinting, and hybrid approaches have been used to navigate the objects in either indoor or outdoor environments. A variety of artificial intelligent methods, e.g., machine learning, neural networks, deep learning, Bayesian networks, fuzzy systems, particle swarm optimization, unsupervised learning techniques, etc., have been proposed for improving the accuracy of localization [9,10]. The genetic algorithm has also been applied for localization [11–13]. Some physical layer localization technologies have been adopted for object tracking in indoor environments; for example, WiFi, RFID, Bluetooth, UWB, Ultrasound, Visible Light, FM radio, Zigbee, LoRa, mobile networks, and Hybrid [14,15]. The 3D Bayesian graphical model has been used for indoor localization systems [16,17]. In the literature [18–21], Modified Particle Swarm Optimization (M-PSO) and AFSA were used to locate the target using the Least Square method. The literature [22–24] used PSO to optimize sensing data such as GPS, inertial sensors, and speedometers for vehicle positioning. In literature [25,26], using PSO, Artificial Neural Network (ANN), and Levenberg Marquardt (LM) training methods, RSSI received by a fixed anchor node from a moving target point was taken as ANN input. The number of hidden layers and the learning rate of the ANN were determined by PSO, and the target localization was estimated by this method. In contrast to the stationary reference node

method for target localization estimation, RSSI received from mobile nodes was used in this study, along with a swarm algorithm to mimic the foraging properties of organisms. The highest points of food sources (RSSI values) are located through the continuous movement of nodes. However, to improve the accuracy of this architecture, it is necessary to increase the number of sensor nodes, which will increase the cost.

This paper discusses the influence of the initial location arrangement and the number of particles in the PSO algorithm used for target localization and tracking. The impact of fixed weight and adaptive weight on improving the PSO algorithm is also considered in the paper. In addition, a Region Segmentation Method (RSM) is proposed in the paper for reducing the localization time investigated. Meanwhile, this paper proposes a Dynamic Individual Selection (DIS) Method for target tracking systems to reduce the computing complexity in the PSO algorithm, examined through simulations.

The rest of the paper is as follows. Section 2 presents the literature review, and the system architecture is introduced in Section 3. Section 4 describes the simulation experiment and results. Finally, a conclusion is given in Section 5.

## 2. RSSI Channel Model and PSO Algorithm

### 2.1. RSSI Channel Model

The RSSI channel model is also known as the propagation path loss model [27–31]. It can be subdivided into the free space propagation model and the log-distance path loss model. The difference between the two is that the free space propagation model is used to estimate the signal intensity received by the receiving end when there is no obstacle or line of sight (LOS) between the transmitter and the receiver. In the free space propagation model, the strength of the signal received by the receiver is inversely proportional to the square of the distance. The most common self-used space model is the Friis free space model [27]. The logarithmic distance path loss model is used to estimate the average power of the receiving end in various environments. Whether it is a theoretical derivation or many research and measurement results, the average power of the received signal will show an exponential attenuation with the increasing distance. In general, the average loss $\overline{PL}(d)$ caused by the path is proportional to the nth power of the distance. In other words, at any distance between the transmitter and the receiver, the average loss at the receiver is expressed by

$$\overline{PL}(d) \propto \left( \frac{d}{d_0} \right)^n \tag{1}$$

The dB value of $\overline{PL}(d)$ is obtained by

$$\overline{PL}[\text{dB}] = \overline{PL}(d_0) + 10n \times log\left( \frac{d}{d_0} \right)[\text{dB}] \tag{2}$$

where $n$ is the path loss exponent, which represents the path loss rate. $d_0$ is the reference distance set by 0.5 m in this study, and $d$ is the distance between the transmitter and the receiver.

At a fixed distance between the transmitter and the receiver, the power loss caused by the propagation path is a random variable. The reason is that, in the natural environment, the interference received by the signal changes with the environment. Therefore, in the simulation analysis, we can only describe this phenomenon with logarithmic normal distribution, so the average power of the path can be expressed as

$$PL(d)[\text{dB}] = \overline{PL}(d) + X_\sigma = \overline{PL}(d_0) + 10n \cdot log\left( \frac{d}{d_0} \right) + X_\sigma \tag{3}$$

The received power can be expressed as

$$P_r(d)[\text{dBm}] = P_t[\text{dBm}] - PL(d)[\text{dB}] \tag{4}$$

where $\overline{PL}(d)$ is the average path loss at the same distance between the transmitter and the receiver. $X_\sigma$ is a zero-mean Gaussian random variable with the standard deviation $\sigma$ set by 9 dB. Many studies have shown that, at the same distance, different receivers have a logarithmic normal distribution of dB values for transmission power loss due to different propagation paths called the Log-normal Shadowing effect. In Equation (4), the received power can be obtained by subtracting the transmission power $P_t$ from the path loss $PL$. Here, if the distance between the receiver and the transmitter is greater than the reference distance, the received power will not be greater than the transmission power. Otherwise, if the distance between the transmitter and the receiver is less than the reference distance, the path loss power obtained from Equation (3) may be minus. If the received power is larger than the transmitting power, the reference distance is 0.5 m in subsequent simulation experiments.

### 2.2. PSO Algorithm

The PSO algorithm is a swarm intelligence-based algorithm proposed by Dr. Russell Kennedy and James Eberhart in 1995 [32,33]. The PSO is derived from the study of bird foraging behavior.

#### 2.2.1. Behavior of the PSO Algorithm

In the PSO algorithm, everyone is regarded as a particle without weight and volume in the D-dimensional search space, and flies at a certain speed [18–22]. Its flight speed is dynamically adjusted by the flight experience of the individual and the whole. If the number of particles is $M$, the position of the $i$th particle can be expressed as $X_i$, the best position experienced by the $i$th particle is $pbest[i]$, the velocity is $V_i$, and the best position among all individuals can be expressed as $gbest$. Therefore, the speed and position can be updated by

$$V_i^t = V_i^{t-1} + c_1 \cdot Rand() \cdot (pbest[i] - X_i) + c_2 \cdot Rand() \cdot (gbest - X_i) \tag{5}$$

and

$$X_i = X_i + V_i \tag{6}$$

respectively. In Equation (5), $V_i^t$ is the velocity of current time of the $i$th particle; $V_i^{t-1}$ is the velocity of last time of the $i$th particle; $c_1$ is the weight of individual experience, which regulates the step length of a particle flying to the individual optimal position. $c_2$ is the weight of the whole experience, adjusting the step size of the particle flying to the whole optimal position. Its value is mostly between [0, 4]. In order to avoid particles leaving the search space, $V_i$ is usually restricted to a certain range, namely $|V_i| \in [0, V_{max}]$, where $V_{\max}$ is the maximum speed. If the search space is in $[-X_{\max}, X_{\max}]$, then $V_{\max} = k\, X_{\max}$, $0.1 \le k \le 1.0$.

#### 2.2.2. Improvement of the PSO Algorithm

In order to improve the convergence ability of the PSO algorithm, the concept of inertia weight was proposed in the literature [33,34]. The method was to add inertia weight $\omega$ into the velocity updating equation, as shown below.

$$V_i^t = \omega \cdot V_i^{t-1} + c_1 \cdot Rand() \cdot (pbest[i] - X_i) + c_2 \cdot Rand() \cdot (gbest - X_i) \tag{7}$$

where $\omega$ is the inertia weight, which can control the influence of the previous velocity on the current velocity [35,36]. When $\omega$ is larger, the influence of the previous velocity is greater, and the global search capability is stronger. When $\omega$ is small, the influence of the previous velocity is small and the local search ability is strong. In the late iteration period, particle mobility should be reduced so that it can explore better solutions in the region. The

adaptive step size and field of view equation proposed in [37] is used here. For example, when the adaptive weight $\omega$ is updated by

$$
\begin{cases}
\omega_t = \omega_{t-1} \times \alpha + \omega_{\min} \\
\alpha = e^{\left(-30 \times \left(\frac{t}{T_{\max}}\right)^s\right)}
\end{cases}
\tag{8}
$$

where $\omega_t$ is the inertia weight of the current iteration; $\omega_{t-1}$ is the inertia weight of the previous iteration; $\omega_{\min}$ is the minimum inertial weight; $s$ is the convergence factor; $t$ is the number of current iterations; $T_{\max}$ is the maximum number of iterations.

## 3. System Architecture

In this section, the methods of target localization and tracking in wireless sensor networks based on the PSO algorithm are described in detail. The RSSI channel model described in Section 2 is used to estimate the distance between the target point and the algorithm individual, and the highest point of the food source (RSSI value) is taken as the estimate point by the individual moving several times. In addition, this paper also discusses the impact of the number of sensors and sensors deployment on the system.

### 3.1. Target Localization Method

In the PSO algorithm, each individual particle will randomly move and change the direction of movement according to its own experience and group experience. Meanwhile, it will compare its own experience with that of other particles to find a better solution. The characteristics of the PSO algorithm cause particles to be not only affected by their own evolution, but to also have the ability to learn and remember inter-group evolution. Furthermore, the particle itself can achieve the best adjustment, and after many moves, find the best position of the fitness value as the estimate point. Figure 1 shows the flow chart of the target localization method of the particle swarm optimization algorithm. In Figure 1, target points were randomly generated in a 100 m square wireless sensor network. The initial position of the particle swarm is divided into random points and regular points to meet the needs of different applications. Its initial velocity generates a two-dimensional matrix in the range of [−1, 1] with random numbers. Then, the particle swarm is compared with each other after multiple movements, and the particle swarm moves towards the direction with the maximum fitness value until the set stopping condition is satisfied. The corresponding position of the last global optimal solution is regarded as the estimate point. In addition, inertia weight is added to the target localization method of the PSO algorithm to improve the efficiency of the algorithm.

#### 3.1.1. Region Segmentation Method

To reduce the number of individuals used at the same time, this paper proposes a region segmentation method (RSM) to improve system performance.

The concept of RSM is to divide a square wireless sensor network with an area of 100 m × 100 m into four equal regions and place an anchor node in the center of each region. The mobile sensor to be arranged is evenly divided into four equal parts and arranged in four regions. The anchor node of the four regions is used to receive the signal strength from the target point to judge the approximate area of the target point, and the boundary is defined $[X_L, X_U, Y_L, Y_U]$, $X_L$ is the lower limit of $X$ axis; $X_U$ is the upper limit of $X$ axis; $Y_L$ is the lower limit of $Y$ axis; $Y_U$ is the upper limit of the $Y$ axis, which limits the individual particle to the range formed by the boundary and then determines which region of the mobile sensor to use for algorithm behavior. Figure 2 is a schematic diagram of RSM region judgment. In Figure 2, the four anchor nodes receive the signal of the target point to obtain four signal strength values, among which the signal strength received by Section 1 is the largest, so it can be judged that the target point may be located in Section 1, and use the Section 1 motion sensor for target location estimation. The boundary is set as [0, 50, 0, 50], which means that the motion sensor can only move within this range. The RSM method

not only reduces the search range of the algorithm, but also reduces the number of mobile sensors used, thereby improving system efficiency.

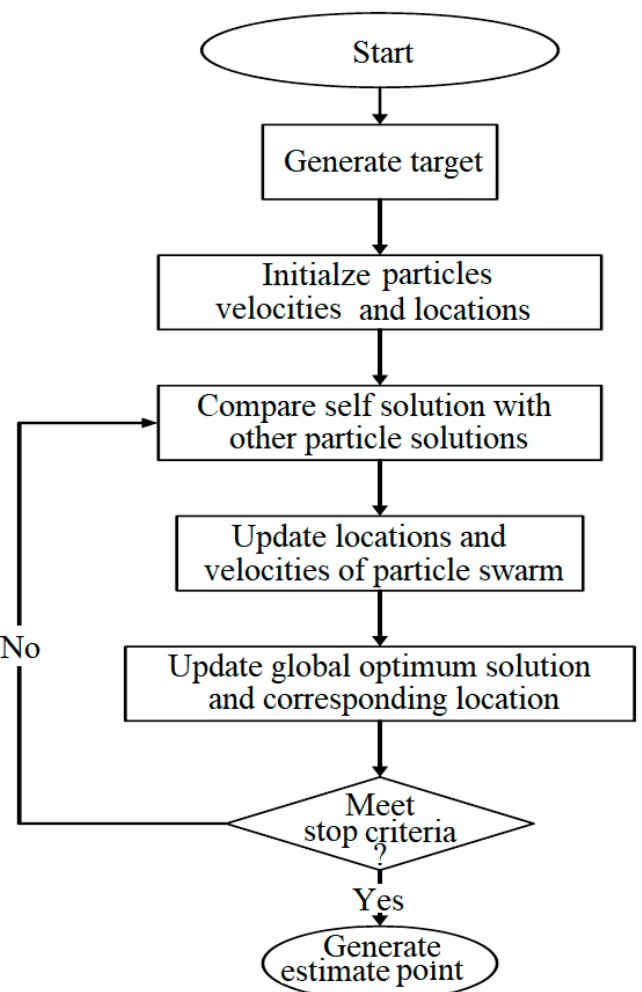

**Figure 1.** Flow chart of the PSO algorithm target localization method.

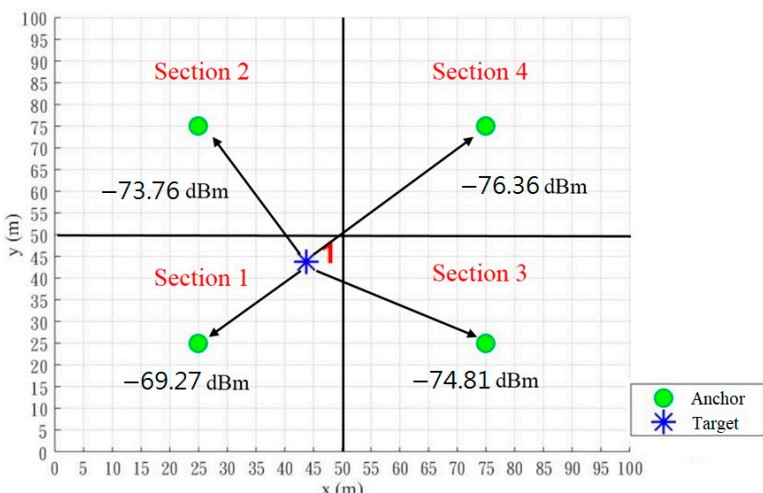

**Figure 2.** Schematic diagram of RSM region judgment.

3.1.2. Random Points Target Localization

In a square wireless sensor network with an area of 100 m × 100 m, it takes less time to set up the network and requires less labor costs to arrange mobile sensors in a random way.

However, the random scattering method may lead to higher node density in some areas or lower node density in some areas, making the efficiency of target location estimation unstable. Therefore, the RSM region segmentation method proposed in this paper uses the method of region division to divide all the mobile sensors to be arranged into four equal parts, and then randomly arrange them in each region. Figure 3 is a schematic diagram of random point target localization with RSM. If the randomly arranged mobile sensor method without RSM is used, 52 mobile sensors will be used for the algorithm behavior, and the search range is a [0, 100] square area. However, with the random placement of mobile sensors method with RSM, the system only needs 13 mobile sensors to perform the algorithm behavior, and the search range is determined by the RSM region judgment described above, which greatly reduces the number of mobile sensors used and reduces the search range of target position estimation. However, due to the reduction in the number of mobile sensors used, the amount of data exchanged between algorithms is also reduced, resulting in a decrease in the number of samples that algorithms can refer to and compare, which may affect the accuracy of the system.

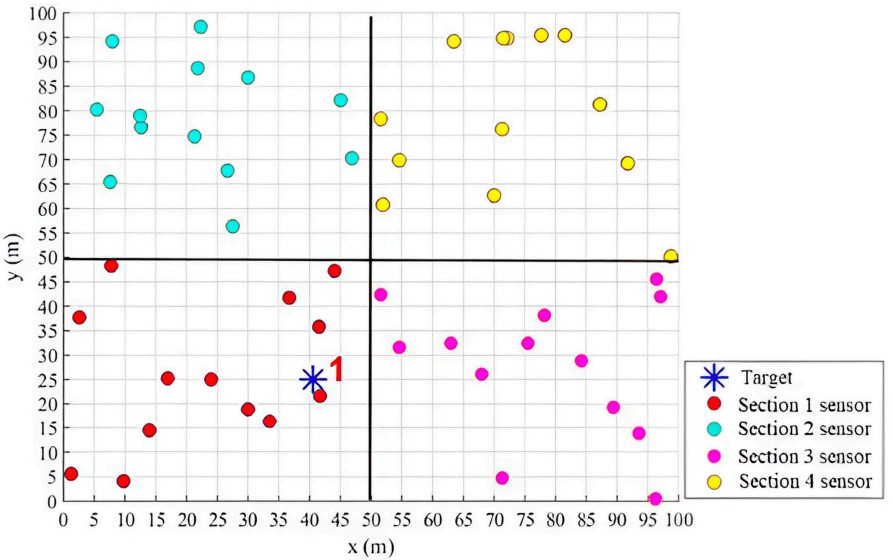

**Figure 3.** Schematic diagram of random point target localization with RSM.

### 3.1.3. Regular Points Target Localization

When the initial position of the mobile sensor is arranged in a fixed way, the topology of the sensor placement is particularly important. In this paper, the mobile sensors with the number of 100, 72, 52, 24, and 12 are arranged in different topologies in the wireless sensor network, and the layout of the mobile sensors is arranged in a symmetrical square as far as possible to avoid the high density of sensors in some areas. Some areas of the sensor density are low. In this paper, RSM is used in the regular point target localization method to improve the efficiency of the target localization system. However, the topology of the moving sensor of the regular point target localization method with RSM is slightly different from that without RSM. However, the arrangement principle is consistent; all abide by the principle of left and right symmetry. Figure 4a–e displays topologies of mobile sensors with RSM.

### 3.2. Target Tracking Method

In this paper, PSO is used to estimate the target position in the target tracking system. However, different from the target positioning system, in the target tracking system, the target point constantly moves in the 100 m × 100 m square wireless sensor network with a specific trajectory. Therefore, how to improve the system's efficiency on the original PSO is one of the issues to be discussed in this section.

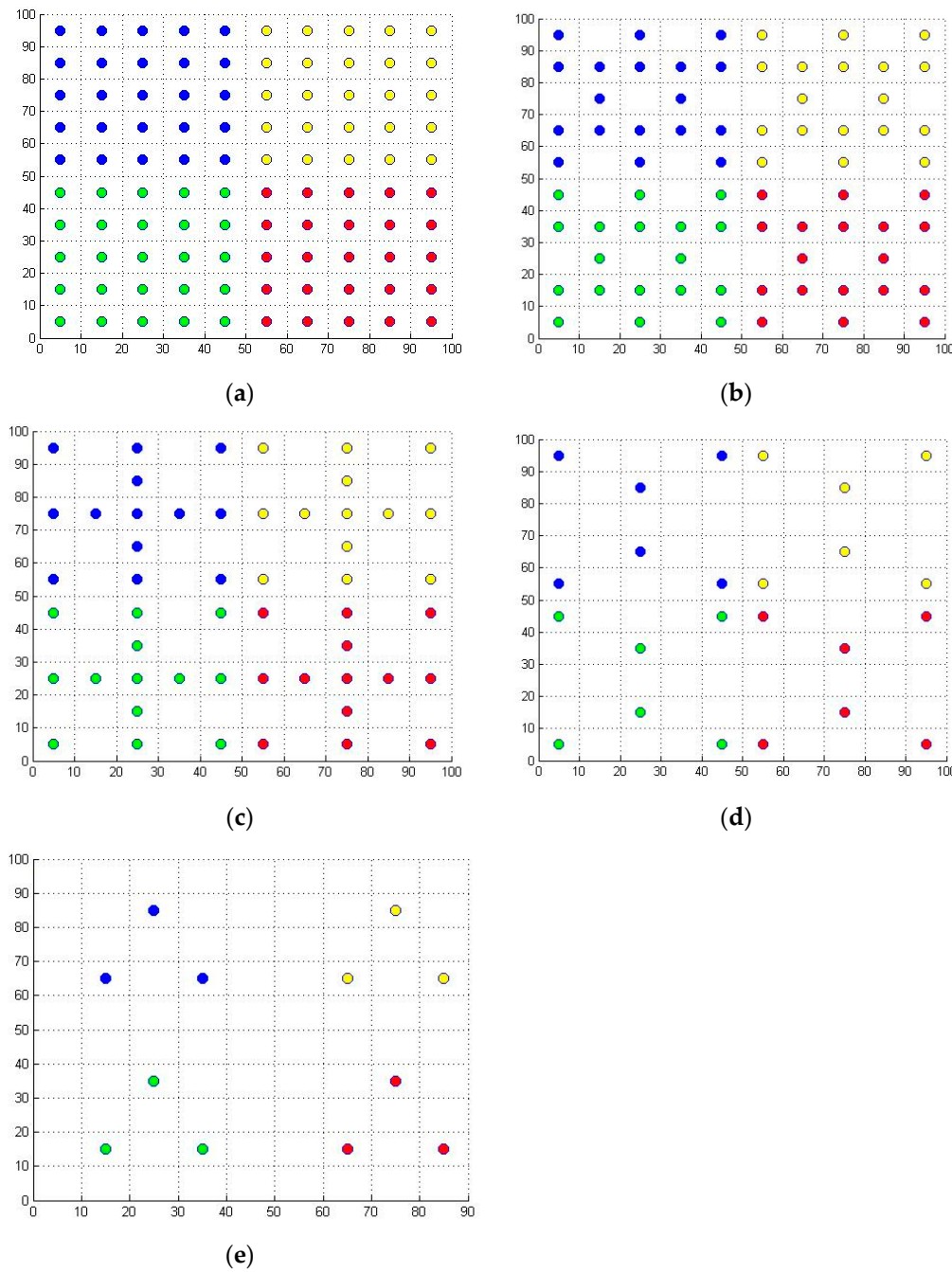

**Figure 4.** Topology of RSM sensors with five different numbers (**a**) 100; (**b**) 72; (**c**) 54; (**d**) 24; (**e**) 12.

The individuals of the PSO algorithm will converge to the region with the largest fitness value. However, due to the moving characteristics of target points in the target tracking system, the individuals of the algorithm will gather around the estimated point (global optimal solution) after finding the estimated point by the algorithm behavior in the last time. By comparing each individual with the global optimal solution after constantly moving, the final global optimal solution will be taken as the estimation point. However, the value of the global optimal solution after the target position estimation in the last time will be much larger than the initial value of the global optimal solution, thus affecting the accuracy of the target position estimation in the current time. Therefore, whether it is fixed-point target tracking or random point target tracking, the mobile sensors will find the estimated point in the last time and return to the initial position. The global optimal

solution is reset to the initial value so that the mobile sensor can estimate the location of the target point at the current time.

In the PSO target tracking method, individuals of the algorithm are placed in a 100 m × 100 m square wireless sensor network in the form of random or fixed scatter points. Through continuous movement and mutual comparison of particle swarms, the particle swarm moves in the direction of higher fitness value. Figure 5 shows the flow chart of PSO target tracking.

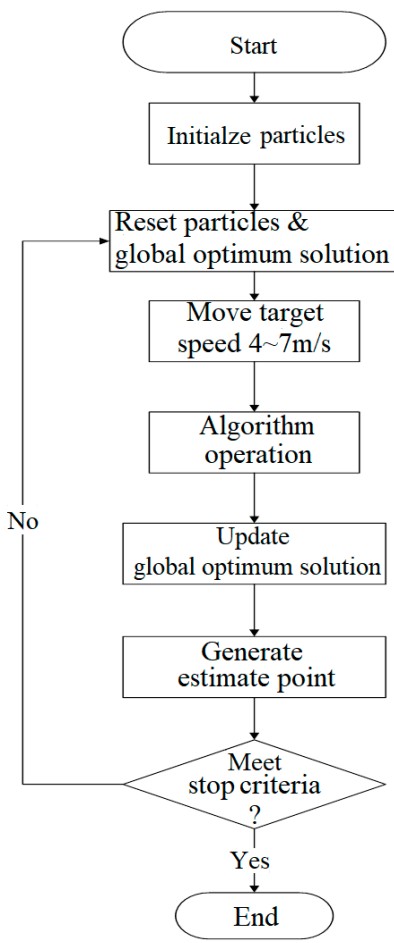

**Figure 5.** Flow chart of PSO target tracking.

### 3.2.1. Target Tracking Method Network Definition

In this paper, two moving modes of target points are designed in the target tracking method, as shown in Figure 6a,b. The moving speed is 4–7 m/s, so the individuals in the algorithm need to find the location of the target point within a limited number of moves. Therefore, the parameter setting of the algorithm is slightly different from the parameters of the algorithm in the target positioning method, such as step size and visual field. How to adjust the parameters of the algorithm so that the individual in the algorithm can quickly find the target point is one of the problems to be discussed in this paper.

### 3.2.2. Individual Movement Restriction in the Algorithm

Due to the moving characteristics of the target point in the target tracking system, the moving rate of the target point is 4–7 m/s. In other words, the individual algorithm needs to move as much as possible within one second and reach the threshold value $\gamma$, which affects the success rate of the system. If the signal intensity received by the individual is greater than or equal to $\gamma$, the search is deemed to have found the target point position and the information such that the current estimated point position and fitness value is

recorded into the recording matrix for subsequent calculation of success rate. Otherwise, the individual can perform the algorithm behavior within one second. If the fitness value of no algorithm individual reaches $\gamma$ within one second, the individual with the highest fitness value among all algorithm individuals will be selected as the estimation point. Figure 7 is the flow chart of individual movement restrictions of the algorithm. Equation (9) is the definition of success rate.

$$\begin{cases} S = find(Record > \gamma) \\ Success\ Rate = \left( \frac{S}{Total} \right) \times 100\% \end{cases} \tag{9}$$

where $S$ is the number of all estimated points reaching $\gamma$. *Record* is the recording matrix, recording the fitness values of all estimated points. *Total* is the number of all estimated points.

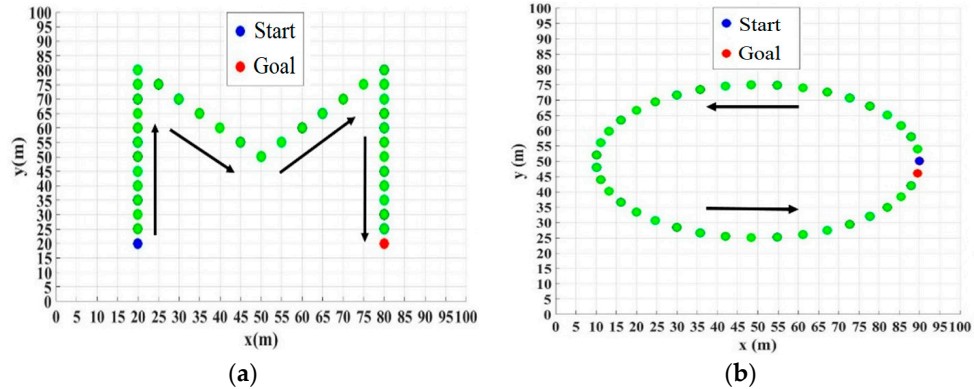

(**a**)                    (**b**)

**Figure 6.** Moving modes of target points for the target tracking method. (**a**) M trace; (**b**) ellipse trace.

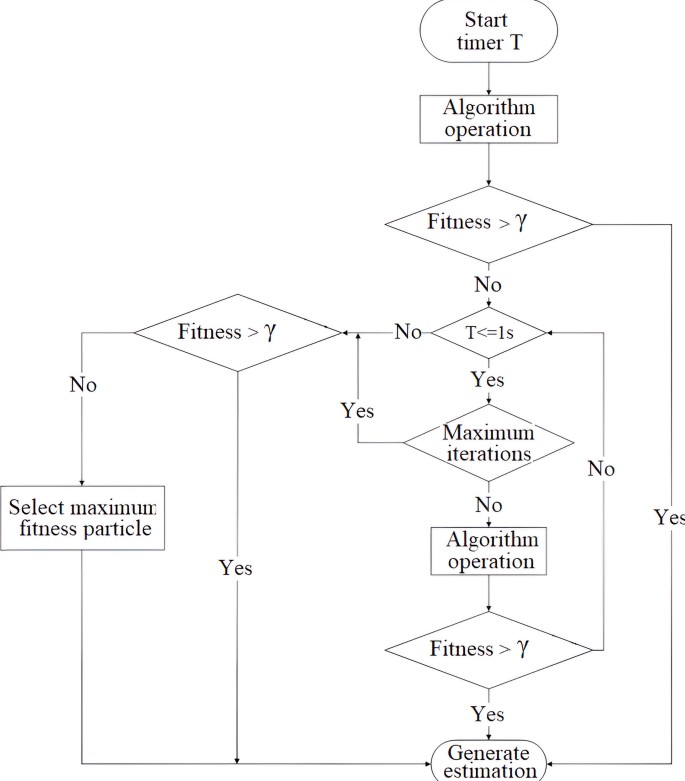

**Figure 7.** Flow chart of individual movement restriction in the algorithm.

### 3.2.3. Dynamic Individual Selection Method

In the PSO target tracking system, no matter whether the region segmentation method (RSM) is used, the target position estimation is carried out by all the individuals in the algorithm or all the individuals in the region, and the number of individual algorithms used is large and the region to be searched by the algorithm is large. In order to reduce the number of individuals used by the algorithm and reduce the search area of the algorithm to improve the efficiency of the target tracking system, this paper proposes a Dynamic Individual Selection (DIS) method in the target tracking system. Given that the moving rate of the target tracking system is 4~7 m/s, in other words, the distance between the current time target point and the previous time target point is not much different, so using DIS method to select the algorithm individual within 20 m from the last time estimate point, a new round of algorithm behavior begins if there is no algorithm individual within 20 m from the last time estimate point. Then, the algorithm individual within 30 m is further selected. If there is no algorithm individual within 30 m, the algorithm behavior is carried out with all algorithm individuals.

In addition, in the DIS method, it selects the surrounding individuals to perform the algorithm behavior based on the position of the estimated point at the last time. Therefore, if the error of the estimated point at the last time is too large, it may cause further error propagation, so that the tracking success rate is not good. Therefore, in the DIS method, if the fitness value of the estimated point in the last time is less than the correction factor $\beta$, the error of the estimated point in the last time is too large, and then all the algorithm individuals are used to perform the algorithm behavior, so that the situation of error propagation can be corrected. Because the DIS method only uses the algorithm individuals around the last estimated point, it not only reduces the search area of the algorithm, but also further reduces the number of individuals used by the algorithm, which improves the efficiency of the algorithm. Figure 8 shows the flow chart of the DIS method.

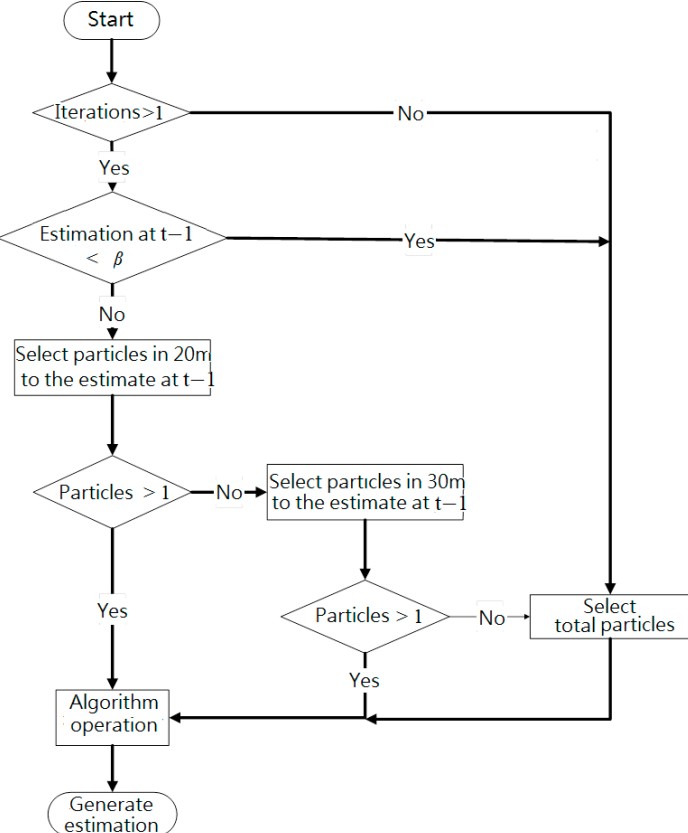

**Figure 8.** Flow chart of DIS method.

## 4. Simulation Results and Discussion

### 4.1. Simulation Environment and Setting

In this paper, the equipped computer and MATLAB software shown in Table 1 are used to conduct simulation experiments. The size of the wireless sensor network is a two-dimensional space of 100 m × 100 m. The PSO algorithm is used for target localization and target tracking. The RSSI value can be obtained by substituting the localization of the algorithm individuals and the target point into the target function. In the same way as the RSSI property previously described, the closer the location is to the target point, the greater the RSSI value is. Therefore, it is assumed that each algorithm individual can know the location of other individuals. After moving many times, the algorithm individual moves towards the region with the largest fitness value, and finally gathers around the global optimal solution. The individual with the largest fitness value is selected as the estimation point.

**Table 1.** Computer equipment.

| Name | Specification |
| --- | --- |
| OS | Windows 7 Enterprise 64 bits |
| CPU | Intel (R) Core (TM) i5-4590 3.30 GHz |
| RAM | 8 GB |
| MATLAB Edition | R2015a |

### 4.2. Simulation Analysis on Target Localization

In the simulation analysis of target localization, this paper can be subdivided into random and regular points according to different sensor scattering modes. In addition, this section analyzes the influence of the methods proposed in this paper on the system efficiency in target localization. To calculate the error between the estimated point and the target point obtained by the PSO target localization, the average error value is calculated by

$$\varepsilon = \frac{1}{N} \sum_{i=1}^{N} \| \hat{x}_i - x_i \| \tag{10}$$

where $N$ is the number of target points. $\hat{x}_i$ is the position of the $i$th estimated point and $x_i$ is the position of the $i$th target point.

In this paper, the RSSI channel model is used to estimate the distance between the algorithm individual and the target point in the simulation. However, since the signal in the real environment varies according to different environments, the larger the standard deviation $\sigma$ of the Gaussian random variable in the RSSI channel model is, the more unstable the signal is; on the contrary, the smaller the standard deviation is, the more stable the signal is. Table 2 shows the RSSI channel model parameters of the simulation experiment.

**Table 2.** RSSI channel model parameters.

| Parameter | Value |
| --- | --- |
| Transmission Power $Pt$ | 2 mW |
| Carrier Frequency $f$ | 2.4 GHz |
| Path Loss Exponent $n$ | 4.5 |
| Reference Distance $d_0$ | 0.5 m |
| Antenna gains $Gt$, $Gr$ | 1 |
| Standard Deviation $\sigma$ | 9 dB |

This section is for the simulation analysis of PSO in target positioning. It will be divided into random and regular points according to the individual's layout methods, and divided into fixed and adaptive weights according to the difference of velocity weight. It will also analyze the influence of the RSM method proposed in this paper on the PSO target

positioning system. In this simulation experiment, the different numbers of algorithm individuals are arranged to explore the application of the proposed method in the PSO target positioning system on the accuracy of target location estimation, system stability, and positioning time. Table 3 shows the PSO target localization analysis method. Table 4 shows the setting of fixed and adaptive weights parameters.

**Table 3.** Analysis modes for target localization method.

| Mode | Condition | | | | |
|---|---|---|---|---|---|
| | **Random Points** | **Regular Points** | **Fixed Weight** | **Adaptive Weight** | **RSM** |
| L1 | ✓ | | ✓ | | |
| L2 | ✓ | | | ✓ | |
| L3 | ✓ | | ✓ | | ✓ |
| L4 | ✓ | | | ✓ | ✓ |
| L5 | | ✓ | ✓ | | |
| L6 | | ✓ | | ✓ | |
| L7 | | ✓ | ✓ | | ✓ |
| L8 | | ✓ | | ✓ | ✓ |

**Table 4.** Fixed and adaptive weights parameters.

| Parameter | Value | |
|---|---|---|
| | **Fixed Weight** | **Adaptive Weight** |
| Network Size | 100 m $\times$ 100 m | |
| Number of executions | 100 | |
| Number of iterations $T_{\max}$ | 100 | |
| Number of particles $M$ | 100, 72, 52, 24, 12 | |
| Number of targets $N$ | 10 | |
| Cognitive Coefficient $c_1$ | 2 | |
| Social Coefficient $c_2$ | 2 | |
| Maximum Velocity $VelMax$ | $0.1 \times networkSize$ | |
| Minimum Velocity $VelMin$ | $-VelMax$ | |
| Velocity Weight $\omega$ | 1 | * |
| Maximum Velocity $\omega_{\max}$ | * | 0.9 |
| Minimum Velocity $\omega_{\min}$ | * | 0.4 |
| Convergence factor $S$ | * | 4 |

* Not applicable.

### 4.2.1. Random Points and Regular Points

The simulation results were analyzed based on the particle arrangement in the algorithm, as shown in Tables 5 and 6. The results show little difference in system efficiency between the regular and random points. However, it could be found that when the number of individuals used in the algorithm was 12, the regular point scattering method was used regardless of whether the RSM method was added. The accuracy and stability of the target position estimation are better than that of the random scattering method. Therefore, it can be concluded that there is no significant difference between the regular scattering method and the random scattering method in system efficiency when the number of algorithm individuals is large. The reason is that every individual in the algorithm can obtain more information when the number of individuals used in the algorithm is large. Therefore, the initial position of the individual algorithm has no great influence on system performance. On the contrary, when the number of individuals in the algorithm is small, everyone can obtain relatively less information from other individuals. Therefore, it is better to avoid missing the global optimal solution by evenly placing the individual algorithm in the sensor network in the form of regular scattering points.

**Table 5.** Average error analysis of random points and regular points in target localization.

| Number of Particles | Regular Points (cm) | | | | | | | | Difference | | | | |
|---|---|---|---|---|---|---|---|---|---|---|---|---|---|
| | L1 | L2 | L3 | L4 | L5 | L6 | L7 | L8 | L1 L5 | L2 L6 | L3 L7 | L4 L8 | Total Average |
| 100 | 3.473 | 0.000 | 10.172 | 0.000 | 3.201 | 0.000 | 10.343 | 0.000 | 8% | 0% | 2% | 0% | |
| 72 | 3.985 | 0.000 | 13.411 | 0.000 | 4.776 | 0.000 | 12.928 | 0.000 | 20% | 0% | 4% | 0% | |
| 54 | 4.958 | 0.000 | 16.409 | 0.000 | 5.576 | 0.000 | 15.459 | 0.000 | 12% | 0% | 6% | 0% | 21% |
| 24 | 8.843 | 0.000 | 26.960 | 2.530 | 9.137 | 0.000 | 25.752 | 7.565 | 3% | 0% | 4% | 199% | |
| 12 | 14.430 | 0.002 | 48.664 | 20.196 | 10.290 | 0.000 | 40.938 | 18.095 | 29% | 100% | 16% | 10% | |

**Table 6.** Average localization time analysis of random points and regular points in target localization.

| Number of Particles | Random Points (s) | | | | Regular Points (s) | | | | Difference | | | | |
|---|---|---|---|---|---|---|---|---|---|---|---|---|---|
| | L1 | L2 | L3 | L4 | L5 | L6 | L7 | L8 | L1 L5 | L2 L6 | L3 L7 | L4 L8 | Total Average |
| 100 | 0.396 | 0.410 | 0.175 | 0.159 | 0.460 | 0.402 | 0.161 | 0.164 | 16% | 2% | 8% | 3% | |
| 72 | 0.317 | 0.311 | 0.134 | 0.142 | 0.314 | 0.342 | 0.138 | 0.132 | 1% | 10% | 3% | 7% | |
| 54 | 0.268 | 0.252 | 0.113 | 0.113 | 0.251 | 0.238 | 0.125 | 0.134 | 6% | 6% | 11% | 19% | 12% |
| 24 | 0.200 | 0.178 | 0.097 | 0.119 | 0.166 | 0.182 | 0.088 | 0.092 | 17% | 2% | 9% | 23% | |
| 12 | 0.111 | 0.128 | 0.094 | 0.100 | 0.194 | 0.138 | 0.082 | 0.103 | 75% | 8% | 13% | 3% | |

#### 4.2.2. The Number of Individuals in the Algorithm

It can be observed from the experimental results that the more individuals in the algorithm, the lower the target positioning error, which indicates that the number of individuals in the algorithm is directly proportional to the stability of the system. The reason is that the more individuals in the algorithm, the more information each individual of the algorithm can obtain at the same time from other individuals, making it easier for the individual of the algorithm to find the global optimal solution. The more individuals there are, the more information the algorithm needs to process, which increases the computational complexity, and thus leads to a longer localization time. Here, analysis mode L1 is taken as an example to simulate the system performance difference between 100 and 12 algorithm individuals, as shown in Figure 9.

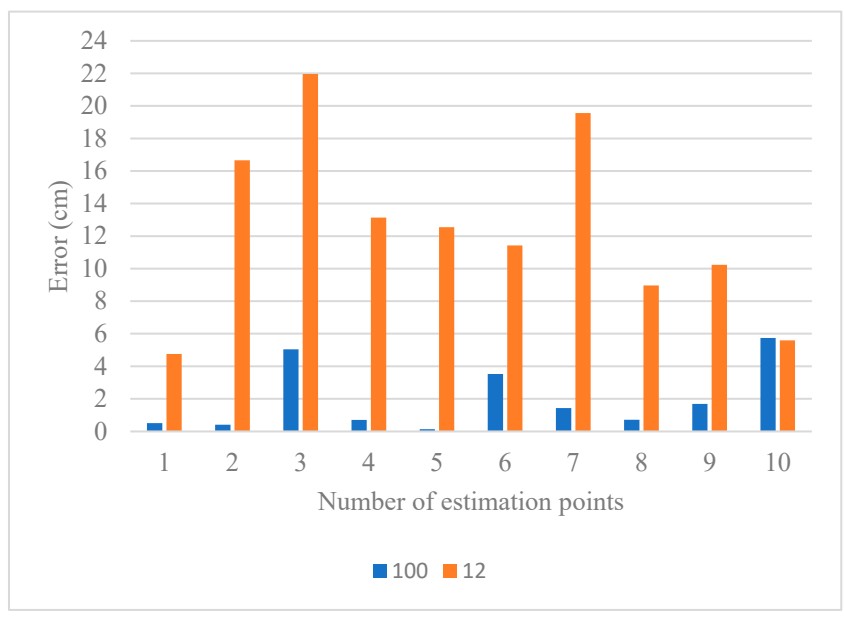

**Figure 9.** Localization error of L1 using 100 and 12 particles in target localization.

#### 4.2.3. Fixed Weight and Adaptive Weight

Taking target localization analysis modes L1 and L2 as examples, Table 7 shows the parameters of L1 and L2. Figure 10 shows the summation of the RSSI of all particles in each

iteration in the process of locating each target point by L1 and L2, and Figure 11 shows the distance between the global optimal solution and the target point in each iteration. It can be seen from the figure that when L1 locates each target point, its RSSI value tends to be flat in the 15th iteration on average, and its RSSI value is around 20.20 dBm, while L2′s RSSI value is still on the rise until the 180th iteration. In addition, as shown in Figure 11, when the number of algorithm individuals is 100, the iteration curve of the algorithm converges to 0 at last, and the error between the global optimal solution and the target point is still present at the 100th iteration of the iteration curve of the algorithm with the number of individuals being 12. The results show that the fixed weight will lead to too much inertia and miss the global optimal solution, while the adaptive weight can adjust the velocity weight according to the search demand in different periods, so that the accuracy of target position estimation is better.

**Table 7.** L1 and L2 parameters.

| Parameter | Value | |
|---|---|---|
| | **L1** | **L2** |
| Network Size | $100 \text{ m} \times 100 \text{ m}$ | |
| Number of executions | 100 | |
| Number of iterations $T_{max}$ | 100 | |
| Number of particles $M$ | 100 | |
| Number of targets $N$ | 10 | |
| Cognitive Coefficient $c_1$ | 2 | |
| Social Coefficient $c_2$ | 2 | 1 |
| Maximum Velocity *VelMax* | $0.1 \times networkSize$ | |
| Minimum Velocity *VelMin* | $-VelMax$ | |
| Velocity Weight $\omega$ | 1 | * |
| Maximum Velocity $\omega_{max}$ | * | 0.9 |
| Minimum Velocity $\omega_{min}$ | * | 0.4 |
| Convergence factor $S$ | * | 4 |

* Not applicable.

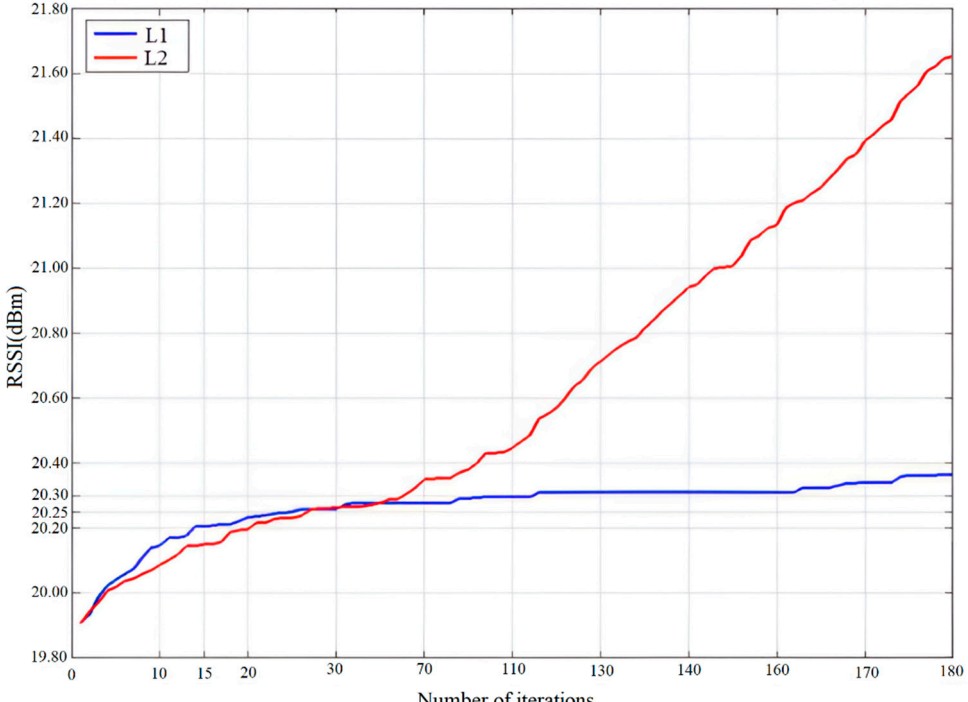

**Figure 10.** The summation of RSSI of all particles in each iteration for L1 and L2.

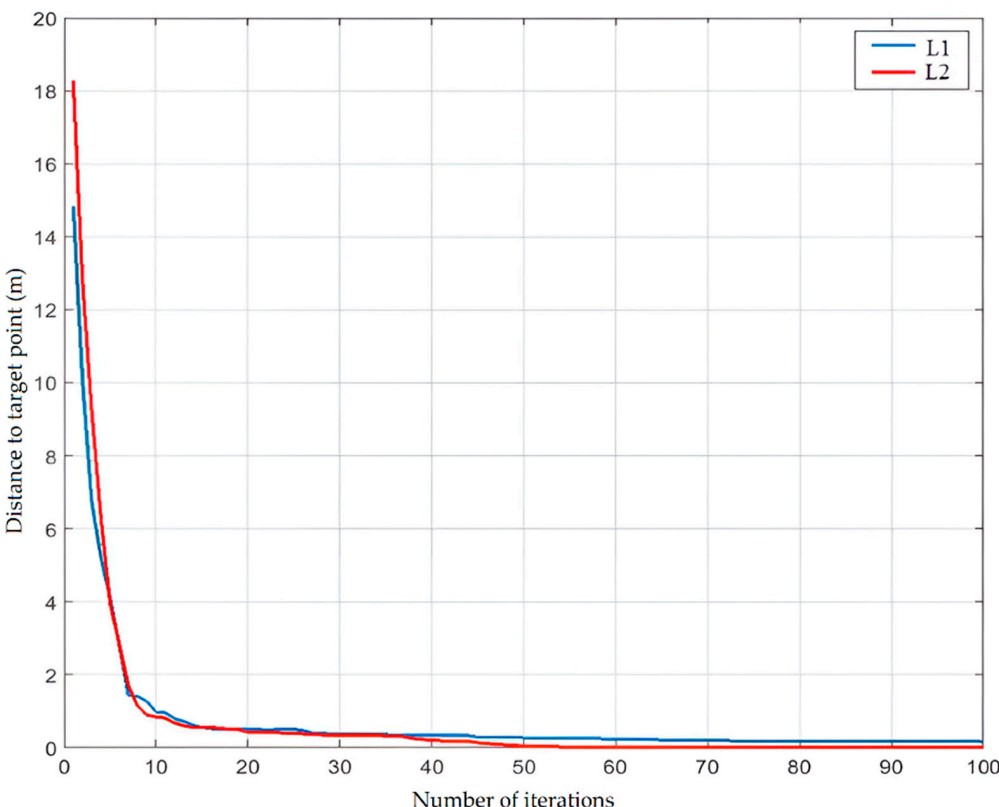

**Figure 11.** The distance between the global optimal solution and the target point for L1 and L2.

#### 4.2.4. RSM Effect

The RSM method reduces the number of individuals used in the algorithm by estimating the approximate area of the target point, and narrows the search range of the individuals in the algorithm to increase the speed of target positioning. We take L1, random points and fixed weight without RSM, and L3, random points and fixed weight with RSM, as examples. The average localization times of L1 and L3 are shown in Table 8. L3 can reduce the average localization time on five different particle numbers by 47.8%. However, Tables 5 and 6 show that although the RSM method improves the speed of target localization, the number of individuals in the RSM method is much smaller than that without RSM, so the information among individuals in the algorithm is also less. If a fixed weight is used in the analysis method with RSM, the error of the algorithm will be further increased. On the contrary, the adaptive weight method makes up for the lack of RSN on accuracy and stability, and retains its advantage of shorting localization time. From Table 5, the total average localization time of different analysis modes with the RMS method for target localization can be reduced by 48.95%.

**Table 8.** The average localization time of analysis modes L1 and L3.

| Number of Particles | L1 (s) | L3 (s) | Difference |
|---|---|---|---|
| 100 | 0.396 | 0.175 | 56% |
| 72 | 0.317 | 0.134 | 58% |
| 54 | 0.268 | 0.113 | 58% |
| 24 | 0.2 | 0.097 | 52% |
| 12 | 0.111 | 0.094 | 15% |

Summarizing the final performance of various analysis methods, as shown in Figure 12, the global optimal solution of the analysis method with adaptive weights is significantly greater than that with fixed weights. For example, L3 uses fixed weights and RSM, and

its final global optimal solution is 20.5 dBm. However, after being replaced with adaptive weights, L4, the final global optimal solution is increased to 20.4 dBm. The results show that the adaptive weight method can improve the accuracy of target position estimation and that the RSM can reduce target localization time.

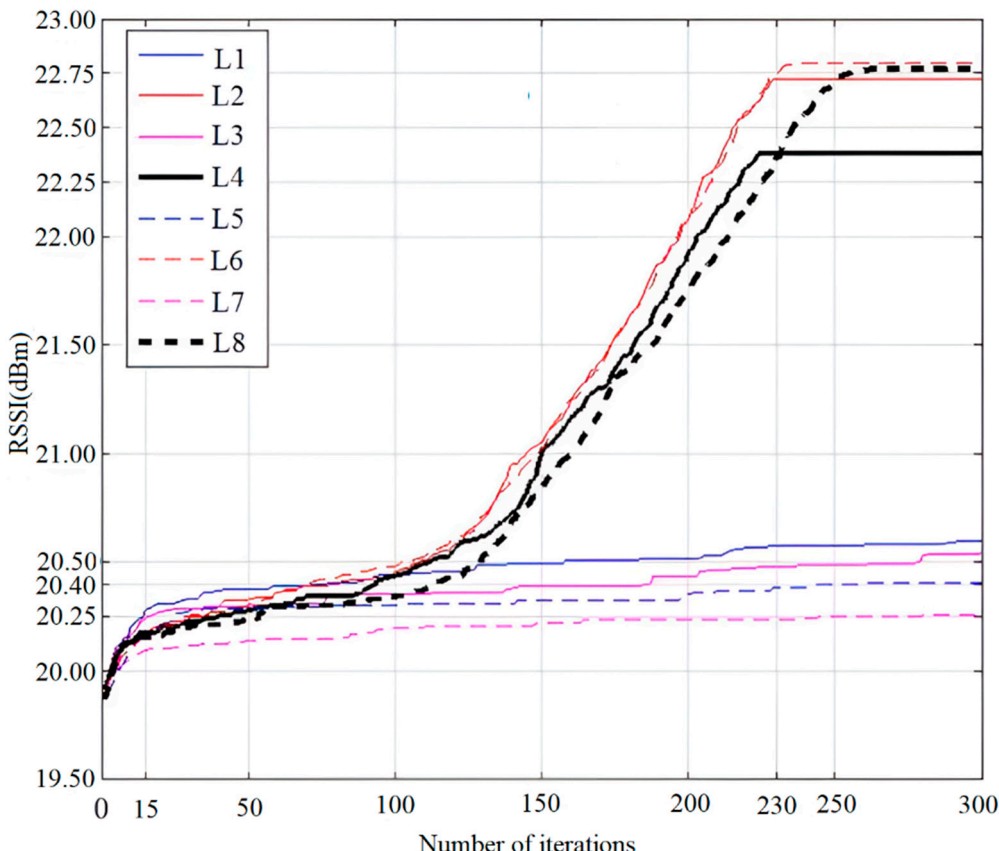

**Figure 12.** The summation of RSSI of all particles in each iteration for L1 to L8.

### 4.3. Simulation Analysis on Target Tracking

The methods proposed in this paper, such as RSM or DIS, are respectively added into the target tracking system. The analysis modes for the target tracking method are shown in Table 9 to analyze their influence on system efficiency. In addition, in the simulation experiment, different numbers of algorithms are placed in the sensor network for the simulation experiment of each analysis method, and the influence of different methods on the system efficiency is discussed. Table 10 shows the simulation parameters of target tracking.

**Table 9.** Analysis modes for target tracking method.

| Mode | Condition | | | |
|---|---|---|---|---|
| | **Random Points** | **Regular Points** | **RSM** | **DIS** |
| T1 | ✓ | | | |
| T2 | ✓ | | ✓ | |
| T3 | ✓ | | | ✓ |
| T4 | ✓ | | ✓ | ✓ |
| T5 | | ✓ | | |
| T6 | | ✓ | ✓ | |
| T7 | | ✓ | | ✓ |
| T8 | | ✓ | ✓ | ✓ |

**Table 10.** Target tracking parameters.

| Parameter | Value |
|---|---|
| Network Size | 100 m × 100 m |
| Number of executions | 100 |
| Number of particles $M$ | 100, 72, 52, 24, 12 |
| Number of targets $N$ | 10 |
| Cognitive Coefficient $c_1$ | 2 |
| Social Coefficient $c_2$ | 2 |
| Maximum Velocity *VelMax* | $0.1 \times networkSize$ |
| Minimum Velocity *VelMin* | $-VelMax$ |
| Maximum Velocity $\omega_{max}$ | 0.9 |
| Minimum Velocity $\omega_{min}$ | 0.4 |
| Convergence factor $S$ | 4 |
| Threshold $\gamma$ | $-6.5$ dBm |
| Correction factor $\beta$ | $-15$ dBm |

### 4.3.1. Random Points and Regular Points

Based on the analysis of the simulation results according to the particle arrangement in the algorithm, it is found that the target tracking efficiency of random points and regular points is similar, as shown in Tables 11 and 12. The average positioning time difference between the two points is 8% on average, and the target tracking success rate between random points and regular points is 1%. The results show that the individual arrangement in the algorithm has no significant influence on target tracking.

**Table 11.** Average tracking time analysis of random points and regular points in target tracking.

| Number of Particles | Random Points (s) | | | | Regular Points (s) | | | | Difference | | | | |
|---|---|---|---|---|---|---|---|---|---|---|---|---|---|
| | T1 | T2 | T3 | T4 | T5 | T6 | T7 | T8 | T1 T5 | T2 T6 | T3 T7 | T4 T8 | Total Average |
| 100 | 0.036 | 0.017 | 0.025 | 0.014 | 0.04 | 0.018 | 0.024 | 0.015 | 10% | 6% | 4% | 7% | |
| 72 | 0.031 | 0.015 | 0.022 | 0.014 | 0.035 | 0.016 | 0.02 | 0.014 | 11% | 6% | 10% | 0% | |
| 54 | 0.03 | 0.015 | 0.019 | 0.015 | 0.031 | 0.014 | 0.017 | 0.015 | 3% | 7% | 12% | 0% | 8% |
| 24 | 0.024 | 0.012 | 0.02 | 0.018 | 0.023 | 0.011 | 0.021 | 0.022 | 4% | 9% | 5% | 18% | |
| 12 | 0.015 | 0.01 | 0.018 | 0.019 | 0.017 | 0.011 | 0.023 | 0.02 | 12% | 9% | 22% | 5% | |

**Table 12.** Target tracking success rate of random points and regular points in target tracking.

| Number of Particles | Random Points | | | | Regular Points | | | | Difference | | | | |
|---|---|---|---|---|---|---|---|---|---|---|---|---|---|
| | T1 | T2 | T3 | T4 | T5 | T6 | T7 | T8 | T1 T5 | T2 T6 | T3 T7 | T4 T8 | Total Average |
| 100 | 100.00% | 100.00% | 100.00% | 99.00% | 100.00% | 100.00% | 100.00% | 97.26% | 0% | 0% | 0% | 2% | |
| 72 | 99.36% | 100.00% | 99.05% | 95.66% | 100.00% | 100.00% | 100.00% | 96.35% | 1% | 0% | 1% | 1% | |
| 54 | 100.00% | 100.00% | 97.74% | 92.92% | 100.00% | 99.17% | 99.17% | 93.78% | 0% | 1% | 1% | 1% | 1% |
| 24 | 95.56% | 97.05% | 84.72% | 71.49% | 99.17% | 98.33% | 84.78% | 75.33% | 4% | 1% | 0% | 5% | |
| 12 | 97.88% | 93.09% | 73.99% | 58.95% | 100.00% | 93.96% | 69.58% | 60.25% | 2% | 1% | 6% | 2% | |

### 4.3.2. DIS Effect

It can be seen from Tables 11 and 12 that the average positioning time of the analysis method with DIS is less than that without DIS. However, when the number of algorithm individuals is small, the success rate of target tracking is greatly reduced by the method with DIS. The reason is that when the number of algorithm individuals is small, less other individual information can be obtained at the same time. Furthermore, the probability of falling into the local optimal solution is increased. At the same time, because DIS is to select the algorithm individual around the last estimated point for algorithm behavior, when the number of algorithm individuals is small, there may be only one algorithm individual around the last estimated point.

### 4.3.3. RSM Effect

The simulation results show that the method with the addition of RSM takes less time to estimate the average position of the target point at time *t* than the method without the addition of RSM. Take random points as an example, T1 without RSM and T2 with RSM, as shown in Tables 13 and 14. It can be seen from Table 13 that RSM can indeed reduce the time of target location estimation by estimating the approximate area of the target point, thus reducing the number of algorithms used and narrowing the search scope of the algorithm. In other words, with the addition of RSM, less time and fewer algorithms can achieve a good tracking success rate. From Table 11, the total average localization time of different analysis modes with the RMS method for target tracking can be reduced by 34.14%. The total average accuracy of target tracking is 93.09%.

**Table 13.** The average tracking time of analysis modes T1 and T2.

| Number of Particles | T1 (s) | T2 (s) | Difference |
|---|---|---|---|
| 100 | 0.036 | 0.017 | 53% |
| 72 | 0.031 | 0.015 | 52% |
| 54 | 0.03 | 0.015 | 50% |
| 24 | 0.024 | 0.012 | 50% |
| 12 | 0.015 | 0.01 | 33% |

**Table 14.** Target tracking success rate of T1 and T2.

| Number of Particles | T1 | T2 | Difference |
|---|---|---|---|
| 100 | 100.00% | 100.00% | 0% |
| 72 | 99.36% | 100.00% | 1% |
| 54 | 100.00% | 100.00% | 0% |
| 24 | 95.56% | 97.05% | 2% |
| 12 | 97.88% | 93.09% | 5% |

## 5. Conclusions

In this paper, PSO is used to study indoor target localization and tracking. The simulation results show that the more algorithms used, the better the accuracy and stability of target location estimation, but the longer the time for target location estimation, so this paper proposes a region segmentation method (RSM). The simulation results show that the proposed RSM method can reduce the number of particles used in the PSO algorithm and improve the speed of positioning and tracking without affecting the accuracy of target localization and tracking. The total average localization time for target localization and tracking with the RSM method can be reduced by 48.95% and 34.14%, respectively, and the average accuracy of target tracking reaches up to 93.09%.

In addition, the DIS method is proposed in the target tracking system to further reduce the number of algorithm individuals used in target location estimation. The simulation results show that the DIS can indeed use fewer individuals than the original number of individuals to achieve a good target tracking effect when the number of individuals is large. However, when the number of algorithms is small, the success rate of target tracking will greatly decrease when the DIS method is used in the target tracking system. By adding RSM and DIS into the target tracking system, the average target tracking success rate of 96.8% can be achieved by using a small number of individuals in the algorithms.

**Author Contributions:** Conceptualization, C.-H.C. and Y.-F.H.; Investigation, C.-H.C., C.-C.L. and Y.-F.H.; Methodology, S.-H.L. and Y.-F.H.; Software, C.-H.C. and C.-C.L.; Supervision, C.-H.C. and Y.-F.H.; Writing—original draft, C.-H.C., S.-H.L., C.-C.L. and Y.-F.H.; Writing—review and editing, S.-H.L., C.-H.C. and Y.-F.H. All authors have read and agreed to the published version of the manuscript.

**Funding:** This research was funded by Ministry of Science and Technology (MOST), R.O.C. grant number MOST 111-2221-E-324-018 and MOST-111-2637-E-150-001.

**Data Availability Statement:** Not applicable.

**Conflicts of Interest:** The authors declare no conflict of interest.

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
