# Peer review of "PSO-Based Target Localization and Tracking in Wireless Sensor Networks"

_electronics, doi:10.3390/electronics12040905_

Round 1

Reviewer 1 Report

This paper has proposed an improved algorithm for WSN using PSO algorithm. There are some questions here:

1.     Section II was “Literature Review”. However, there were no related works introduced in this section. The authors just provided the basic description of “RSSI Channel Mode” and “PSO”.

2.     Almost all the references were not lastest ones. Some related works should be provided and analyzed in recent three years.

3.     Why did RSM divide the WSN into four sections? If more scetions were divided, could it reduce the required data and search range to determine the random target location?

4.     In sub-section 4.2.4, the illustration should be provided to analyze the impact from RSM on target location in detail.

5.     This paper should provide the comparison and analysis between the proposed algorithms and the baseline algorithms.

Author Response

Response to Reviewer 1 Comments

Point 1:  Section II was “Literature Review”. However, there were no related works introduced in this section. The authors just provided the basic description of “RSSI Channel Mode” and “PSO”.

Response 1: Thank you very much for this comment. The title of Section 2 in the revised version has been changed to “RSSI Channel Model and PSO Algorithm” to match the content.

Point 2: Almost all the references were not lastest ones. Some related works should be provided and analyzed in recent three years.

Response 2: Thank you very much for this comment. In the revised paper, 11 related works published in the recent three years have been included in the reference.

[A1] Billa; I. Shayea; A. Alhammadi; Q. Abdullah; M. Roslee. An Overview of Indoor Localization Technologies: Toward IoT Navigation Services. In Proceedings of 2020 IEEE 5th International Symposium on Telecommunication Technologies (ISTT), Shah Alam, Malaysia, 2020; pp. 76-81, https://doi.org/10.1109/ISTT50966.2020.9279369.

[A2] Alhammadi A; Hashim F; A Rasid MohdF; Alraih S. A three-dimensional pattern recognition localization system based on a Bayesian graphical model. International Journal of Distributed Sensor Networks, 2020,16,9. https://doi.org/10.1177/1550147719884893

[A3] K. A. Kordi; A. Alhammadi; M. Roslee; M. Y. Alias; Q. Abdullah. A Review on Wireless Emerging IoT Indoor Localization. In Proceedings of 2020 IEEE 5th International Symposium on Telecommunication Technologies (ISTT), Shah Alam, Malaysia, 2020; pp. 82-87. https://doi.org/10.1109/ISTT50966.2020.9279386.

[A4] S.J. Hayward; K. van Lopik; C. Hinde; A.A. West. A Survey of Indoor Location Technologies, Techniques and Applications in Industry. Internet of Things,2022,20,100608. https://doi.org/10.1016/j.iot.2022.100608.

[A5] Jiazhi Ni; Fusang Zhang; Jie Xiong; Qiang Huang; Zhaoxin Chang; Junqi Ma; BinBin Xie; Pengsen Wang; Guangyu Bian; Xin Li; Chang Liu. Experience: pushing indoor localization from laboratory to the wild. In Proceedings of the 28th Annual International Conference on Mobile Computing And Networking (MobiCom '22), New York, NY, USA, 2022; pp.147–157. https://doi.org/10.1145/3495243.3560546

[A6] Arigye W; Pu Q; Zhou M; Khalid W; Tahir MJ. RSSI Fingerprint Height Based Empirical Model Prediction for Smart Indoor Localization. Sensors (Basel), 2022, 22, 23:9054. https://doi.org/10.3390/s22239054.

[A7] Xia, B.; Liu, T.; Ding, T. et al. An Improved PSO Localization Algorithm for UWB Sensor Networks. Wireless Pers Commun.,2021, 117, pp. 2207–2223. https://doi.org/10.1007/s11277-020-07968-x

[A8] Nagireddy, Vyshnavi; Parwekar, Pritee;Mishra, Tusar Kanti. Velocity adaptation based PSO for localization in wireless sensor networks. Evolutionary Intelligence, 2021, 14, 1. https://doi.org/10.1007/s12065-018-0170-4.

[A9] Yang, Q. A new localization method based on improved particle swarm optimization for wireless sensor networks. IET Soft. 2022, 16, 3, pp. 251– 258. https://doi.org/10.1049/sfw2.12027

[A10] Sun D; Wei E; Ma Z; Wu C; Xu S. Optimized CNNs to Indoor Localization through BLE Sensors Using Improved PSO. Sensors (Basel), 2021, 21, 6 :1995. https://doi.org/10.3390/s21061995.

[A11] Saeed Doostali; Mohammad Khalily-Dermany. A multi-hop PSO based localization algorithm for wireless sensor networks. Soft Computing Journal, 2021, 8, 1, pp. 58-69. https://doi.org/10.22052/8.1.58

Point 3: Why did RSM divide the WSN into four sections? If more sections were divided, could it reduce the required data and search range to determine the random target location?

Response 3: Thank you very much for this comment. The reason why the RSM divides a square wireless sensor network into four equal regions is to form four smaller square regions so that the anchor in them can be deployed easily to reduce calculation during processing. From the simulation results, although the RSM method improves the speed of the target location, the number of related sensors in the RSM method is much less than that without RSM. Therefore, less information can be used between the individuals in the algorithm. If the number of individuals in the split regions is lower than a limitation then the error of the algorithm will be increased. Therefore, if the RSM splits the square wireless sensor network into more than four regions, the limitation of the number of individuals in the region maintaining a good performance should be considered. However, the question about the number of split regions in RSM is very interesting for further study in the future.

Point 4:   In sub-section 4.2.4, the illustration should be provided to analyze the impact from RSM on target location in detail.

Response 4: Thank you very much for this comment. The detail analysis about the impact of RSM on target location has been added in the sub-section 4.2.4.

“The RSM method reduces the number of individuals used in algorithm by estimating the approximate area of the target point, and narrows the search range of the individuals in algorithm to increase the speed of target positioning. We take L1, random points and fixed weight without RSM, and L3, random points and fixed weight with RSM, as an example. The average location time of L1 and L3 is shown in Table 8. L3 can reduce the average location time on five different particle numbers by 47.8%. However, Tables 5 and 6 show that although the RSM method improves the speed of target location, the number of individuals in the RSM method is much smaller than that without RSM, so the information among individuals in the algorithm is also less. If fixed weight is used in the analysis method with RSM, the error of the algorithm will be further increased. On the contrary, the adaptive weight method makes up for the lack of RSN on accuracy and stability, and retains its advantage of shorting location time. From Table 5, the total average location time of different analysis modes with RMS method for target location can be reduced by 48.95%.

Table 8. The average location time of analysis modes L1 and L3.

Number of Particles

L1 (sec)

L3 (sec)

Difference

100

0.396

0.175

56%

72

0.317

0.134

58%

54

0.268

0.113

58%

24

0.2

0.097

52%

12

0.111

0.094

15%

Point 5:   This paper should provide the comparison and analysis between the proposed algorithms and the baseline algorithms.

Response 5: Thank you very much for this comment. This study investigates 8 different analysis modes based on PSO algorithm for target localization and tracking. L1, random points and fixed weight, is similar to the conventional PSO algorithm. The performance comparison between analysis modes has been highlighted in the revised edition.

“Summarizing the final performance of various analysis methods, as shown in Figure 12, the global optimal solution of the analysis method with adaptive weights is significantly greater than that with fixed weights. For example, L3 uses fixed weights and RSM, and its final global optimal solution is 40dBm. However, after replaced with adaptive weights, L4, the final global optimal solution is increased to 237.5dBm. The results show that the adaptive weight method can improve the accuracy of target position estimation and the RSM can reduce the target location time.”

Reviewer 2 Report

In this paper, the authors demonstrate the capabilities of the particle swarm optimization (PSO) algorithm with the RSSI channel model in indoor target determination and tracking within a wireless sensor network environment. A special emphasis is placed on the impact of different combinations of random or regular points, and fixed or adaptive weights in PSO algorithms on localization performance and target tracking, where the WSN had a different number of nodes (12, 24, 52, 72, and 100). Moreover, the region segmentation method (RSM) is proposed to reduce the number of particles used in the PSO algorithm. The theoretical background related to RSSI Channel Model and PSO algorithm as well as the description of the methods of target location and tracking in wireless sensor networks based on PSO algorithm are presented in a clear and consistent manner. The simulation results show that the proposed RSM method can improve the speed of positioning and tracking without affecting the accuracy of target localization and tracking. Also, analyzing these results, it can be clearly concluded that the use of Dynamic Individual Selection (DIS) method in the target tracking system can significantly reduce the number of algorithm individuals used in target location estimation.

All in all, the paper contains a satisfactory scientific contribution that is necessary for publication in a journal.

Author Response

Response to Reviewer 2 Comments

Point : In this paper, the authors demonstrate the capabilities of the particle swarm optimization (PSO) algorithm with the RSSI channel model in indoor target determination and tracking within a wireless sensor network environment. A special emphasis is placed on the impact of different combinations of random or regular points, and fixed or adaptive weights in PSO algorithms on localization performance and target tracking, where the WSN had a different number of nodes (12, 24, 52, 72, and 100). Moreover, the region segmentation method (RSM) is proposed to reduce the number of particles used in the PSO algorithm. The theoretical background related to RSSI Channel Model and PSO algorithm as well as the description of the methods of target location and tracking in wireless sensor networks based on PSO algorithm are presented in a clear and consistent manner. The simulation results show that the proposed RSM method can improve the speed of positioning and tracking without affecting the accuracy of target localization and tracking. Also, analyzing these results, it can be clearly concluded that the use of Dynamic Individual Selection (DIS) method in the target tracking system can significantly reduce the number of algorithm individuals used in target location estimation.

All in all, the paper contains a satisfactory scientific contribution that is necessary for publication in a journal.

Response: Thank you very much for this comment.

Reviewer 3 Report

In this paper, the authors propose a region segment method (RSM) to accelerate the target positioning process. In addition, a dynamic individual selection (DIS) method has been proposed to further reduce the computational complexity.

This proposed approaches are well explained, and the results show significant improvement in terms of convergence speed.

However, the illustration of experimental results needs to be treated with extra effort. The entire Section 4 (simulation results and discussion) needs to be justified and revised. Major concerns are as follows:

1. Table 5, 6, 10, 12 have NO units. What are the units of error? In addition, "random points" condition only applies to L1-4 according to Table 3, why L5-8 appears in "random points" condition column in Table 5, 6, 10? Please treat the tables and results with EXTRA care.

2. You use RSSI (abbreviation not explained in your manuscript, I assume it is "received signal strength indicator") as your adaption value, how did you come up with the plots in Figure 10 and Figure 12? If the "average RSSI of each iteration" is the summation of RSSIs at each sensor,  that does NOT make any sense even if you have 100 particles. For example, 100dBm is 10000kW, that is more than the summation of 100 high performance electric vehicle engine output. However the "average RSSI" in your plots can even achieve 250dBm.

3. The simulation parameters in Table 2 are not practical. When path loss gradient ("path loss exponent" in Table 2) 4.5, with 2.4GHz radio frequency, your signal strength could lose 135dB within the first 10 meters of transmission. You will barely receive anything if your transmitter and receiver are separated over 5 meters.

Some minor concerns are listed below. Please go through your manuscript for other issues and typos:

1. In Equation (2)-(4), the unit of path loss shall be marked as "dB" instead of "dBm". "dBm" is a unit of power not ratio.

2. In page 4 line 155, V^{t}_{i} and V^{t}_{i-1} are declared as velocity and speed, respectively. Speed is not the same as velocity, speed is just the rate while velocity is "rate and direction". Looking at your Equation 6, they should be velocity. Please also make X and V bold (or add arrowhead above them) because they are vectors.

Author Response

Response to Reviewer 3 Comments

 In this paper, the authors propose a region segment method (RSM) to accelerate the target positioning process. In addition, a dynamic individual selection (DIS) method has been proposed to further reduce the computational complexity.

This proposed approaches are well explained, and the results show significant improvement in terms of convergence speed.

However, the illustration of experimental results needs to be treated with extra effort. The entire Section 4 (simulation results and discussion) needs to be justified and revised. Major concerns are as follows:

Point 1: Table 5, 6, 10, 12 have NO units. What are the units of error? In addition, "random points" condition only applies to L1-4 according to Table 3, why L5-8 appears in "random points" condition column in Table 5, 6, 10? Please treat the tables and results with EXTRA care.

Response 1: Thank you very much for this comment. Tables 5, 6,10,11 have been modified, shown below, to avoid confusion in the revised edition.

“   Table 5. Average error analysis of random points and regular points in target location.

Number

of Particles

Random Points (cm)

Regular Points (cm)

Difference

L1

L2

L3

L4

L5

L6

L7

L8

L1 L5

L2 L6

L3 L7

L4 L8

Total

Average

100

3.473

0.000

10.172

0.000

3.201

0.000

10.343

0.000

8%

0%

2%

0%

21%

72

3.985

0.000

13.411

0.000

4.776

0.000

12.928

0.000

20%

0%

4%

0%

54

4.958

0.000

16.409

0.000

5.576

0.000

15.459

0.000

12%

0%

6%

0%

24

8.843

0.000

26.960

2.530

9.137

0.000

25.752

7.565

3%

0%

4%

199%

12

14.430

0.002

48.664

20.196

10.290

0.000

40.938

18.095

29%

100%

16%

10%

Table 6. Average location time analysis of random points and regular points in target location.

Number

of Particles

Random Points (sec)

Regular Points (sec)

Difference

L1

L2

L3

L4

L5

L6

L7

L8

L1 L5

L2 L6

L3 L7

L4 L8

Total

Average

100

0.396

0.410

0.175

0.159

0.460

0.402

0.161

0.164

16%

2%

8%

3%

12%

72

0.317

0.311

0.134

0.142

0.314

0.342

0.138

0.132

1%

10%

3%

7%

54

0.268

0.252

0.113

0.113

0.251

0.238

0.125

0.134

6%

6%

11%

19%

24

0.200

0.178

0.097

0.119

0.166

0.182

0.088

0.092

17%

2%

9%

23%

12

0.111

0.128

0.094

0.100

0.194

0.138

0.082

0.103

75%

8%

13%

3%

Table 11. Average tracking time analysis of random points and regular points in target tracking.

Number

of Particles

Random Points (sec)

Regular Points (sec)

Difference

T1

T2

T3

T4

T5

T6

T7

T8

T1 T5

T2 T6

T3 T7

T4 T8

Total

Average

100

0.036

0.017

0.025

0.014

0.04

0.018

0.024

0.015

10%

6%

4%

7%

8%

72

0.031

0.015

0.022

0.014

0.035

0.016

0.02

0.014

11%

6%

10%

0%

54

0.03

0.015

0.019

0.015

0.031

0.014

0.017

0.015

3%

7%

12%

0%

24

0.024

0.012

0.02

0.018

0.023

0.011

0.021

0.022

4%

9%

5%

18%

12

0.015

0.01

0.018

0.019

0.017

0.011

0.023

0.02

12%

9%

22%

5%

Table 12. Target tracking success rate of random points and regular points in target tracking.

Number

of Particles

Random Points

Regular Points

Difference

T1

T2

T3

T4

T5

T6

T7

T8

T1 T5

T2 T6

T3 T7

T4 T8

Total

Average

100

100.00%

100.00%

100.00%

99.00%

100.00%

100.00%

100.00%

97.26%

0%

0%

0%

2%

1%

72

99.36%

100.00%

99.05%

95.66%

100.00%

100.00%

100.00%

96.35%

1%

0%

1%

1%

54

100.00%

100.00%

97.74%

92.92%

100.00%

99.17%

99.17%

93.78%

0%

1%

1%

1%

24

95.56%

97.05%

84.72%

71.49%

99.17%

98.33%

84.78%

75.33%

4%

1%

0%

5%

12

97.88%

93.09%

73.99%

58.95%

100.00%

93.96%

69.58%

60.25%

2%

1%

6%

2%

Point 2: You use RSSI (abbreviation not explained in your manuscript, I assume it is "received signal strength indicator") as your adaption value, how did you come up with the plots in Figure 10 and Figure 12? If the "average RSSI of each iteration" is the summation of RSSIs at each sensor, that does NOT make any sense even if you have 100 particles. For example, 100dBm is 10000kW, that is more than the summation of 100 high performance electric vehicle engine output. However the "average RSSI" in your plots can even achieve 250dBm.

Response 2: Thank you very much for this comment.  The full name of RSSI, the received signal strength indication, has been added in the revised version. The RSSI in Figure 10 and Figure 12 is the summation of RSSI of all particles, and the number of particles is 100. Therefore, when the tracking position is near the target, the RSSI of the particle will be near 3dBm. Thus, the total RSSI of all particles can be greater than 20 dBm. The wrong scale values for RSSI have been corrected in the revised version.

Figure 10. The summation of RSSI of all particles in each iteration for L1 and L2.

Figure 12. The summation of RSSI of all particles in each iteration for L1 to L8.

Point 3: The simulation parameters in Table 2 are not practical. When path loss gradient ("path loss exponent" in Table 2) 4.5, with 2.4GHz radio frequency, your signal strength could lose 135dB within the first 10 meters of transmission. You will barely receive anything if your transmitter and receiver are separated over 5 meters.

Response 3: Thank you very much for this comment. The path loss exponent is set to 4.5 due to the environment considered for simulation is obstructed in building. We assume the path loss at d0 is a free space loss, thus the path loss at d0=0.5m is 34dB. If d=10m, form Equation (3), we can obtain the path loss about PL(d=10m)= 34+45 log(10/0.5)=34+45x1.3=92.5dB. Indeed, the received signal is very weak when the particle is more than 10m far away from the target point. However, that particle will move to the position closer and closer to the target point through the operation of the PSO algorithm.

Some minor concerns are listed below. Please go through your manuscript for other issues and typos:

Point 4: In Equation (2)-(4), the unit of path loss shall be marked as "dB" instead of "dBm". "dBm" is a unit of power not ratio.

Response 4: Thank you very much for this comment. The error unit has been corrected in the revised version.

(2)

(3)

(4)

Point 5: In page 4 line 155, V^{t}_{i} and V^{t}_{i-1} are declared as velocity and speed, respectively. Speed is not the same as velocity, speed is just the rate while velocity is "rate and direction". Looking at your Equation 6, they should be velocity. Please also make X and V bold (or add arrowhead above them) because they are vectors.

Response 5: Thank you very much for this comment. The errors have been corrected in the revised edition.

In the PSO algorithm, everyone is regarded as a particle without weight and volume in the D-dimensional search space, and flies at a certain speed [18]-[22]. Its flight speed is dynamically adjusted by the flight experience of the individual and the whole. If the number of particles is M, the position of the ith particle can be expressed as Xi, the best position experienced by the ith particle is pbest[i], the velocity is Vi, and the best position among all individuals can be expressed as gbest. Therefore, the speed and position can be updated by

,

(5)

and

,

(6)

respectively. In Equation (5), is the velocity of current time of ith particle; is the velocity of last time of ith particle; c1 is the weight of individual experience, which regulates the step length of particle flying to the individual optimal position. c2 is the weight of the whole experience, adjusting the step size of the particle flying to the whole optimal position. Its value is mostly between [0,4]. In order to avoid particles leaving the search space, Vi is usually restricted to a certain range, namely , where Vmax is the maximum speed. If the search space is in [-Xmax, Xmax], then Vmax = k Xmax,0.1 k 1.0.

Reviewer 4 Report

- In the abstract, the authors mentioned "target localization and tracking is investigated when the number of sensors is 12, 24, 52, 72, and 100." The number of sensors and particles is different. Sensors mean the RSSI sources, whereas the number of particles is related to the PSO algorithm. Please check on this.

          - The final outcome of this study should be stated in the abstract, for example, system accuracy, localization error, etc...

In the introduction, The papers' contribution should be clearly stated. I suggest rewriting the paper's contributions on several points. 

In section 2, the authors should add more related works rather than the description of channel models and PSO. Several works should be added as follows:   An overview of indoor localization technologies: Toward IoT navigation services", " A review on wireless emerging IoT indoor localization" and "A three-dimensional pattern recognition localization system based on a Bayesian graphical model".

In section 3, How you measured RSSI in Figure 2? You have only one transmitter ( the target) and four receivers (anchor nodes), but the RSSIs are very high since there are very large distances between the transmitter and receivers that are against path loss theory. Please check on this.

         -On page 09, line 282, "in Figure 6. (a) and (b)." there are two full stops.

- In section 4, page 11, line 356, please write clearly the symbols. 

          - Tables 5 and 6 are confusing, especially the second row, as there are different modes( L1 - L8). Why you consdiered (L1,L5),  (L2,L6), (L3,L7) and (L4,L8) as same aand have same analysis. 

          - In table 7, Number of executions =1, why you did not increase the number of executions? Is it have the same result?

          - Figure 12, why RSSI in positive dBm, it is not correct. The RSSI should be always negative. please study more on the propagation model.

          - You should state the unit of average tracking time analysis in Tables 5  and 10.

          - Figure 9 - 13 should improve their qualities. 

          - Tables 5 and 6 are confusing, especially the second row, as there are different modes( T1 - T8). Why you consdiered (T1,T5),  (T2,T6), (T3,T7) and (T4,T8) as same aand have same analysis.

         - Why do you only consider T1 and T2 analysis  Tables 12 and 13? 

         - I did not see any comparison analysis with state-of-the-art works. You may compare it with one or two recent works.  

Author Response

Response to Reviewer 4 Comments

Point 1: In the abstract, the authors mentioned "target localization and tracking is investigated when the number of sensors is 12, 24, 52, 72, and 100." The number of sensors and particles is different. Sensors mean the RSSI sources, whereas the number of particles is related to the PSO algorithm. Please check on this.

Response 1: Thank you very much for this comment. The unclear statement has been amended in the revised edition.

 “The performance of 8 different method combinations of random or regular points, fixed or adaptive weights, and the region segmentation method (RSM) proposed in this paper for target localization and tracking is investigated for the number of particles in the PSO algorithm with 12, 24, 52, 72, and 100. “

Point 2: The final outcome of this study should be stated in the abstract, for example, system accuracy, localization error, etc...

Response 2: Thank you very much for this comment. The related outcomes have been added in the abstract in the revised version.

“The simulation results show that the proposed RSM method can reduce the number of particles used in the PSO algorithm and improve the speed of positioning and tracking without affecting the accuracy of target localization and tracking. The total average location time for target location and tracking with RSM method can be reduced by 48.95% and 34.14% respectively, and the average accuracy of target tracking is up to 93.09%”

Point 3: In the introduction, the papers' contribution should be clearly stated. I suggest rewriting the paper's contributions on several points. 

Response 3: Thank you very much for this comment. The paper’s contribution has been described in Section I introduction.

“The paper’s contributions include: 1. This paper discusses the influence of the initial location arrangement and the number of particles in the PSO algorithm used for target location and tracking. 2.The impact of fixed weight and adaptive weight improving the PSO algorithm is also considered in the paper. 3. In addition, a Region Segmentation Method (RSM) is proposed in the paper for reducing the location time is investigated. 4. Meanwhile, this paper proposed a Dynamic Individual Selection (DIS) Method for target tracking system to reduce the computing complexity in the PSO algorithm is examined through simulations.”

Point 4: In section 2, the authors should add more related works rather than the description of channel models and PSO. Several works should be added as follows:   An overview of indoor localization technologies: Toward IoT navigation services", " A review on wireless emerging IoT indoor localization" and "A three-dimensional pattern recognition localization system based on a Bayesian graphical model".

Response 4: Thank you very much for this comment. The title of Section 2 in the revised version has been changed to “RSSI Channel Model and PSO Algorithm” to match the content. And a brief overview of indoor localization technologies has been added in Section I introduction.

“A variety of indoor localization technologies have been proposed in the literature. Localization techniques on signal processing methodology, such as proximity sensing, lateration, angulation, dead reckoning, fingerprinting, and hybrid approaches have been used to navigate the objects in either indoor or outdoor environments. A variety of artificial intelligent methods, e.g. machine learning, neural network, deep learning, Bayesian net-work, fuzzy system, particle swarm optimization, unsupervised learning technique et.al., have been proposed for improving the accuracy of localization [9][10]. Genetic algorithm also has been applied for localization [11]-[13]. Some physical layer localization technologies are adopted for object tracking in indoor environments, for example, WiFi, RFID, Bluetooth, UWB, Ultrasound, Visible Light, FM radio, Zigbee, LoRa, mobile networks, and Hybrid [14][15]. The 3D Bayesian graphical model has been used for indoor localization systems[16][17].”

Point 5: In section 3, How you measured RSSI in Figure 2? You have only one transmitter ( the target) and four receivers (anchor nodes), but the RSSIs are very high since there are very large distances between the transmitter and receivers that are against path loss theory. Please check on this.

Response 5: Thank you very much for this comment. Figure 2 is used to demonstrated the RSM method only and its more detail description has been added in the revised version.

“Figure 2 is a schematic diagram of RSM region judgment. In Figure2, the four anchor nodes receive the signal of the target point to obtain four signal strength values, among which the signal strength received by Section 1 is the largest, so it can be judged that the target point may be located in Section 1, and use Section 1 motion sensor for target location estimation. And the boundary is set as, which means that the motion sensor can only move within this range. The RSM method not only reduces the search range of the algorithm, but also reduces the number of mobile sensors used, thereby improving the system efficiency. “

Point 6: On page 09, line 282, "in Figure 6. (a) and (b)." there are two full stops. In section 4, page 11, line 356, please write clearly the symbols. 

Response 6: Thank you very much for this comment. The correction has done in the revised paper.

“as shown in Figure 6 (a) and (b).”  in Line 301.

“ is the position of the ith estimated point and  is the position of the ith target point.” in Line 376.

Point 7: Tables 5 and 6 are confusing, especially the second row, as there are different modes( L1 - L8). Why you consdiered (L1,L5),  (L2,L6), (L3,L7) and (L4,L8) as same as and have same analysis. 

Response 7: Thank you very much for this comment. Tables 5 and 6 have been modified, shown below, to avoid confusion in the revised edition.

“   Table 5. Average error analysis of random points and regular points in target location.

Number

of Particles

Random Points (cm)

Regular Points (cm)

Difference

L1

L2

L3

L4

L5

L6

L7

L8

L1 L5

L2 L6

L3 L7

L4 L8

Total

Average

100

3.473

0.000

10.172

0.000

3.201

0.000

10.343

0.000

8%

0%

2%

0%

21%

72

3.985

0.000

13.411

0.000

4.776

0.000

12.928

0.000

20%

0%

4%

0%

54

4.958

0.000

16.409

0.000

5.576

0.000

15.459

0.000

12%

0%

6%

0%

24

8.843

0.000

26.960

2.530

9.137

0.000

25.752

7.565

3%

0%

4%

199%

12

14.430

0.002

48.664

20.196

10.290

0.000

40.938

18.095

29%

100%

16%

10%

Table 6. Average location time analysis of random points and regular points in target location.

Number

of Particles

Random Points (sec)

Regular Points (sec)

Difference

L1

L2

L3

L4

L5

L6

L7

L8

L1 L5

L2 L6

L3 L7

L4 L8

Total

Average

100

0.396

0.410

0.175

0.159

0.460

0.402

0.161

0.164

16%

2%

8%

3%

12%

72

0.317

0.311

0.134

0.142

0.314

0.342

0.138

0.132

1%

10%

3%

7%

54

0.268

0.252

0.113

0.113

0.251

0.238

0.125

0.134

6%

6%

11%

19%

24

0.200

0.178

0.097

0.119

0.166

0.182

0.088

0.092

17%

2%

9%

23%

12

0.111

0.128

0.094

0.100

0.194

0.138

0.082

0.103

75%

8%

13%

3%

Point 8: In table 7, Number of executions =1, why you did not increase the number of executions? Is it have the same result?

Response 8: Thank you very much for this comment. The Number of executions =1 in Table 7 is a typo and has been corrected to 100 in the revised edition.

Point 9: Figure 12, why RSSI in positive dBm, it is not correct. The RSSI should be always negative. please study more on the propagation model.

Response 9: Thank you very much for this comment. The RSSI in Figure 10 and Figure 12 is the summation of RSSIs of all particles, and the number of particles is 100. Therefore, when the tracking position is near the target, the RSSI of the particle will be near 3dBm. Thus, the total RSSI of all particles can be greater than 20 dBm. The wrong scale values for RSSI have been corrected in the revised version.

Figure 10. The summation of RSSI of all particles in each iteration for L1 and L2.

Figure 12. The summation of RSSI of all particles in each iteration for L1 to L8.

Point 10: You should state the unit of average tracking time analysis in Tables 5 and 10.

Response 10: Thank you very much for this comment. The unit (sec) has been added in Tables 5 and 10.

Point 11: Figure 9 - 13 should improve their qualities. 

Response 11: Thank you very much for this comment. The quality of the Figures has been improved in the revised paper.

Point 12: Tables 5 and 6 are confusing, especially the second row, as there are different modes( T1 - T8). Why you consdiered (T1,T5),  (T2,T6), (T3,T7) and (T4,T8) as same aand have same analysis.

Response 12: Thank you very much for this comment. Tables 10 and 11 have been modified, shown below, to avoid confusion in the revised edition.

Table 11. Average tracking time analysis of random points and regular points in target tracking.

Number

of Particles

Random Points

Regular Points

Difference

T1

T2

T3

T4

T5

T6

T7

T8

T1 T5

T2 T6

T3 T7

T4 T8

Total

Average

100

0.036

0.017

0.025

0.014

0.04

0.018

0.024

0.015

10%

6%

4%

7%

8%

72

0.031

0.015

0.022

0.014

0.035

0.016

0.02

0.014

11%

6%

10%

0%

54

0.03

0.015

0.019

0.015

0.031

0.014

0.017

0.015

3%

7%

12%

0%

24

0.024

0.012

0.02

0.018

0.023

0.011

0.021

0.022

4%

9%

5%

18%

12

0.015

0.01

0.018

0.019

0.017

0.011

0.023

0.02

12%

9%

22%

5%

Table 12. Target tracking success rate of random points and regular points in target tracking.

Number

of Particles

Random Points

Regular Points

Difference

T1

T2

T3

T4

T5

T6

T7

T8

T1 T5

T2 T6

T3 T7

T4 T8

Total

Average

100

100.00%

100.00%

100.00%

99.00%

100.00%

100.00%

100.00%

97.26%

0%

0%

0%

2%

1%

72

99.36%

100.00%

99.05%

95.66%

100.00%

100.00%

100.00%

96.35%

1%

0%

1%

1%

54

100.00%

100.00%

97.74%

92.92%

100.00%

99.17%

99.17%

93.78%

0%

1%

1%

1%

24

95.56%

97.05%

84.72%

71.49%

99.17%

98.33%

84.78%

75.33%

4%

1%

0%

5%

12

97.88%

93.09%

73.99%

58.95%

100.00%

93.96%

69.58%

60.25%

2%

1%

6%

2%

Point 13: Why do you only consider T1 and T2 analysis Tables 12 and 13? 

Response 13: Thank you very much for this comment. To investigate the RSM effect, analysis mode T1 and T2 are adopted as an example, due to T1 without RSM and T2 with RSM. A clear description has been added in the revised paper.

“Take random points as an example, T1 without RSM and T2 with RSM, as shown in Table 13 and Table 14. “

Point 14: I did not see any comparison analysis with state-of-the-art works. You may compare it with one or two recent works.  

Response 14: Thank you very much for this comment. This study investigates 8 different analysis modes based on the PSO algorithm for target localization and tracking. L1, random points, and fixed weight are similar to the conventional PSO algorithm, and L2, random points, and adaptive weight are improved PSO algorithms. The proposed RSM is used in L3 and L4 under random point conditions. The performance comparison between the 8 analysis modes has been examined and discussed in the paper. The comparison analysis with state-of-the-art works using other technologies will be considered in future study.

Reviewer 5 Report

I am responding to MDPI request for me to provide you with a report in connection with the submitted manuscript “PSO Based Target Localization and Tracking in Wireless Sensor Networks” by Shu-Hung Lee, Chia-Hsin Cheng, Chien-Chih Lin , and Yung-Fa Huang. I am pleased to do so.

In this article, the authors have used a particle swarm optimization algorithm with the RSSI channel model for indoor target localization and tracking. They have studied, the performance of different combinations of random or regular points and fixed or adaptive weights in PSO algorithms for target localization and tracking is investigated for a few numbers of sensors (12, 24, 52, 72, and 100). Furthermore, they claimed that the region segmentation method (RSM) is proposed to reduce the number of particles used in the PSO algorithm. The simulation results show that the proposed RSM method can improve the speed of positioning and tracking without affecting the accuracy of target localization.

Their research topic is very interesting, the presentation is very good and their results appear to be correct. The only comments are the following:

1)     What the “RSSI” stands for? It is not mentioned in the abstract section. Please include it there. I think that there is also the same problem with the “RSS”

2)     In the introduction section, could the authors mention any genetic algorithms   related to “fast” optimization techniques for research in electronics? e.g.”SoftwareX 10, 100355 (2019)”. Is there any comparison between the forementioned algorithms?

3)     I would appreciate if they could include more highlights in “conclusions” section.

Author Response

Response to Reviewer 5 Comments

Point 1: In this article, the authors have used a particle swarm optimization algorithm with the RSSI channel model for indoor target localization and tracking. They have studied, the performance of different combinations of random or regular points and fixed or adaptive weights in PSO algorithms for target localization and tracking is investigated for a few numbers of sensors (12, 24, 52, 72, and 100). Furthermore, they claimed that the region segmentation method (RSM) is proposed to reduce the number of particles used in the PSO algorithm. The simulation results show that the proposed RSM method can improve the speed of positioning and tracking without affecting the accuracy of target localization.

Response 1: Thank you very much for this comment.

Point 2: Their research topic is very interesting, the presentation is very good and their results appear to be correct.

Response 2: Thank you very much for this comment.

Point 3: The only comments are the following:

What the “RSSI” stands for? It is not mentioned in the abstract section. Please include it there. I think that there is also the same problem with the “RSS”

Response 3: Thank you very much for this comment. RSS is received signal strength, and RSSI stands for received signal strength Indication, an indicator of RSS. The full name of RSSI has been added in the abstract section in the revised version.

Point 4: In the introduction section, could the authors mention any genetic algorithms   related to “fast” optimization techniques for research in electronics? e.g.”SoftwareX 10, 100355 (2019)”. Is there any comparison between the forementioned algorithms?

Response 4: Thank you very much for this comment. The reference about genetic algorithm has been included in the revised edition.

  1. L. G. Tsoulos; V. Stavrou; N. E. Mastorakis; D. Tsalikakis. GenConstraint: A programming tool for constraint optimization problems. SoftwareX, 2019, 10,100355. https://doi.org/10.1016/j.softx.2019.100355.

Point 5:  I would appreciate if they could include more highlights in “conclusions” section.

Response 5: Thank you very much for this comment. The results of this paper have been highlighted in “conclusions” in the revised edition.

“The simulation results show that the proposed RSM method can reduce the number of particles used in the PSO algorithm and improve the speed of positioning and tracking without affecting the accuracy of target localization and tracking. The total average location time for target location and tracking with RSM method can be reduced by 48.95% and 34.14% respectively, and the average accuracy of target tracking is up to 93.09%.”

Round 2

Reviewer 1 Report

This paper was revised enough. 

Author Response

Point: This paper was revised enough. 

Response: Thank you very much for your kind comment.

Reviewer 3 Report

Most of comments have been addressed. No additional concern about the manuscript.

For my last minor concern, please see below.

In your response 3, I have no doubt about the calculation as 4.5 gradient for the intense environment. Since the large path-loss gradient value is used in simulation, please revise your Figure 2 accordingly to make it more realistic (currently 100m*100m yet the RSSI can be high as -30dBm).

Author Response

Point: Most of comments have been addressed. No additional concern about the manuscript.

For my last minor concern, please see below.

In your response 3, I have no doubt about the calculation as 4.5 gradient for the intense environment. Since the large path-loss gradient value is used in simulation, please revise your Figure 2 accordingly to make it more realistic (currently 100m*100m yet the RSSI can be high as -30dBm).

Response: Thank you very much for this comment. The unrealistic values in Figure 2 has been corrected in the latest revised version.

Figure 2. schematic diagram of RSM region judgment.

Reviewer 4 Report

Thanks for addressing my concerns, but there is one concern that has not been addressed correctly. In Figures 10 and 12, the RSSI can not be positive as I mentioned in the previous review. There is an issue with the simulation, please check this again. The authors must address this issue before final acceptance. 

Author Response

Point: Thanks for addressing my concerns, but there is one concern that has not been addressed correctly. In Figures 10 and 12, the RSSI can not be positive as I mentioned in the previous review. There is an issue with the simulation, please check this again. The authors must address this issue before final acceptance. 

Response: Thank you very much for this comment. The RSSI in Figure 10 and Figure 12 is the summation of RSSI of all particles, and the number of particles is 100. In our simulation, some particles will be near to the target after the target tracking iterations. For examples, when the particles move to near the target about 1 meter, the received power Pr of the particles will be about 0.088 mWatts. The RSSI is approximated -9dBm. However, because there are total 100 particles in the simulation, the summation of RSSI of all particles will be 8.8 mWatts. So, the summation of RSSI can be obtained by 9.4 dBm. Therefore, when the tracking position of the particles is very near the target, the RSSI of the particles will be approximated to 3dBm. Thus, the total RSSI of all particles can be greater than 20 dBm. 
